# SCALABLE PRETRAINING OF RETRIEVAL MODELS

## ABSTRACT

Modern retrieval models are typically pretrained on masked language modeling (MLM) or causal language modeling (CLM) tasks, where retrieval capabilities emerge only incidentally. Achieving state-of-the-art performance requires fine-tuning these models on supervised query-document pairs, which are expensive to curate and sparse for specialized domains. We propose a scalable pretraining approach that directly targets retrieval tasks using a fully self-supervised pipeline. By leveraging web-scale text corpora, we construct retrieval pairs without manual annotations. Our method employs a novel contrastive objective that aligns prefix embeddings generated by a causal transformer with suffix embeddings from an anti-causal transformer. This enables the model to learn fine-grained associations between queries and their completions, analogous to the next-token prediction paradigm in generative models. Our results demonstrate the viability of this method, suggesting that it can bridge the data scarcity gap in information retrieval and imbue retrieval models with the zero-shot reasoning capabilities typically reserved for generative language models.

## 1 INTRODUCTION

Generative language models have the capability to follow complex instructions. The extreme power of these models is derived from extensive unsupervised pretraining that expends nearly all of their training budget. Modern retrieval models do the opposite. While they are pretrained, they gain nearly all of their performance during an expensive secondary phase of *supervised* training. Retrieval models do not benefit from pretraining as directly as generative models currently do. This is because, unlike generative models which pretrain on a next-token-prediction task that reflects the model's inference time use, the MLM and CLM pretraining tasks of retrieval models have no resemblance to their test-time use case.

The lack of correspondence between pretraining and retrieval tasks means that achieving strong retrieval performance requires training on large corpora of curated query-document pairs. This has several bad consequences.

- While large corpora exist for generic benchmark domains, very few pair-wise supervised retrieval datasets of significant size exist for specialized domains like code, law, medicine, or sequence data beyond natural language.
- Retrieval models have weaker instruction following and reasoning capabilities than generative models. This is because generative models are pretrained on trillion token datasets that dwarf the multi-million document corpora used for supervised retrieval training.
- Even if large training corpora were available, retrieval models fail to exploit the efficiencies and parallelism that transformers were built for. Supervised document training only produces one loss per document, while generative model pretraining produces $n$ predictions and losses from a single forward pass on $n$ tokens.

To overcome these issues, we develop a retrieval analog of the standard next-token prediction loss used to train modern generative models (see Figure 1). In our pipeline, we load a single training sequence (e.g., random web text) and pass it through a transformer to produce output embeddings at each token position. We use a *causal* mask so that each token only looks upstream. Rather than interpret these embeddings as next-token predictors (as in a generative model), we interpret these embeddings as representation vectors for each prefix in the sample. Simultaneously, we pass the same token sequence through a transformer block with *anti-causal* masking, yielding another set of

embeddings that represent each suffix. Finally, we compute a contrastive loss that forces each prefix embedding to match with the embedding of its corresponding suffix.

**Motivation and relation to generative models.**
A typical generative causal language model (CLM) produces vectors that correlate with the next *token* in a document. After pretraining, we expect a generative model to answer questions and follow instructions, provided the right prompt engineering is used. Given the prompt "`Question: Who was the first president of the United States? Answer:`" a token prediction model outputs an embedding vector that retrieves the token "George" from the vocabulary. The model can then be instruction tuned so the Question/Answer prompt template is not needed, however, the fundamental capability is imbued by pretraining.

Likewise, our proposed **P**refix-**S**uffix **L**anguage **M**odel (PSLM) (depicted in Figure 1) produces vectors that correlate with the entire *completion* of a document rather than a single word. Given a prompt of the form "`Question: Provide a biography of the first president of the United States? Answer:`" the last embedding vector of the model should align with (and retrieve) an article about George Washington. Instruction tuning can then be used to reformat the model to respond to queries directly, although our focus here is mostly on pretraining.

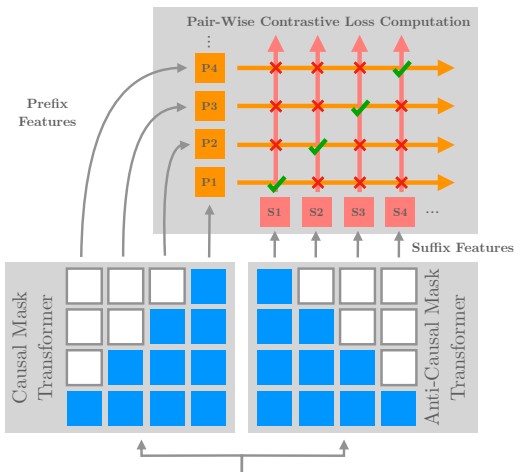

Figure 1: **A schematic diagram of the self-supervised pretraining objective.** A transformer with causal attention is used to extract an embedding vector for every prefix in a training sample. At the same time, a transformer with anti-causal attention masking embeds every suffix. Finally, a contrastive loss is computed that forces each prefix vector to match the correct suffix, while not matching incorrect suffixes.

**Summary of results and contributions**

To validate our **P**refix-**S**uffix **L**anguage **M**odel (PSLM) approach, we train a pair of retrieval models at the 160m and 1.4B parameter scale on 400B tokens [1]. We make several observations.

- We compare our proposed PSLM approach to MLM and CLM pretraining baselines using 100B tokens of identical training data for each, finding superior sample complexity for PSLM (Figure 2).

- We measure the instruction following ability of our PSLM models on the IFEval benchmark, which matches complex multi-part instructions to responses. With only pretraining, our 160M and 1.4B models outperform strong finetuned baselines, including GritLM-7b (a variant of Mistral-7b (Table 2). On the MTEB embedding evaluation (which contains many tasks in addition to retrieval) a pretrained PSLM model scores comparably to other open-source models (Table 1).

- We study each stage of the training pipeline. We observe that PSLM pretraining scales well across model sizes and token counts, with performance being noticeably non-saturated in our training runs. We apply a standard finetuning recipe to our model, observing that our model responds well to finetuning, but off-the-shelf training schemes with hard negatives seem to perform poorly. Inference time testing shows that, like CLM models, our models respond well to prompt engineering.

---

[1]We use the descriptors "-160m" and "-1.4b" for our PSLM models to note their relationship to the transformer block specification from the Pythia-160M and Pythia-1.4B models on which we base the architecture Biderman et al. (2023). Due to the lack of decoder head and a change to LLaMA-2's tokenizer, which has a smaller vocabulary size, the actual trainable parameter counts come out to 110M and 1.27B respectively.

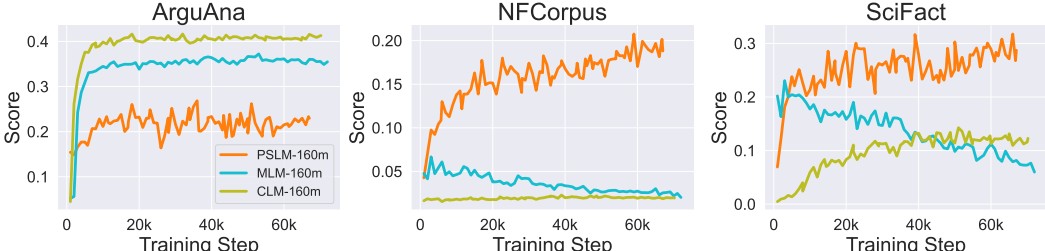

Figure 2: **Comparison of zero-shot retrieval performance as a function of Phase-1 training on FineWeb-100B.** We see that PSLM enjoys steady performance improvements as a function of pretraining steps on two of our three validation datasets, vastly outperforming the MLM and CLM objective under identical conditions. While we report the aggregated performance over all three tasks in Figure 5 across all three phases of training, we show the decomposition here including the peculiar ArguAna "counter-argument retrieval" dataset. We observe that certain tasks contained in benchmark suites like MTEB seem to respond less strongly to both invested compute as well as the alignment of the pretraining objective with the actual task of retrieval.

## 2 PREFIX-SUFFIX LANGUAGE MODELING (PSLM)

Conventional finetuning approaches rely on labeled datasets, such as web questions and answers, as supervision for the contrastive training objective. Contrastive training on this data produces only one vector per query/document, and only one loss term per example. Our approach leverages the natural parallelism enjoyed by the transformer architecture to produce feature representations for all prefixes simultaneously (and all suffixes via anti-causal attention), enabling retrieval training with the scalability and efficacy that we enjoy from generative model training. We dub this objective "Prefix Suffix Language Modeling" primarily based on its relationship with existing architectures and objectives[2].

### 2.1 ANTI-CAUSAL ATTENTION

A single forward pass through a transformer with causal attention yields token embeddings that represent prefixes (all previous tokens) of the text sequence. Similarly, a forward pass with anti-causal attention generates token embeddings corresponding to suffixes (all following tokens). This approach allows us to efficiently obtain prefix embeddings of shape $(n, d)$ and suffix embeddings of shape $(n, d)$ for a sequence of length $n$ in one forward pass per attention type. We then apply a standard contrastive learning objective, InfoNCE (Oord et al., 2018), on these embeddings to train the model.

We perform anti-causal attention with a simple trick: We reverse both the token sequence and positional encoding, then pass it through a causal Flash Attention. This achieves anti-causal attention with the efficiency of specialized causal implementations.

For example, consider a batch of sequences:

$$[\textit{Cat sat on a mat, The cat jumped over the dog}]$$

Passing this batch through the causal prefix model yields token embeddings (assuming a word-based tokenizer):

$$[[P_1^1, P_2^1, P_3^1, P_4^1, P_5^1], [P_1^2, P_2^2, P_3^2, P_4^2, P_5^2, P_6^2]]$$

We then remove the last token embedding of each sequence in the batch (i.e., $P_5^1, P_6^2$) and flatten the list to obtain all prefix embeddings $[P_1, P_2, P_3, \ldots, P_9]$. We also pass the tokens through the model with an anti-causal mask to grab the suffix embeddings $[S_1, S_2, S_3, \ldots, S_9]$. Computing the dot product between these prefix and suffix embeddings yields a similarity matrix where the diagonal elements represent positive pairs. This operation is visualized in Figure 4.

---

[2]We acknowledge that our architecture and loss does not constitute a "language model" according to the classical definition one might find in Jurafsky & Martin (2025) for example, but we find the term illustrative nonetheless.

## 2.2 DISTRIBUTED CONTRASTIVE LOSS

The standard InfoNCE contrastive loss (Oord et al., 2018) computes a similarity matrix of size $n \times n$ by taking $n$ query features and $n$ document features, computing their dot product. Formally, given a query $q$ and a document $d$, both are independently encoded using the same model $f_\theta$, parameterized by $\theta$. The relevance score $s(q, d)$ is then computed as:

$$s(q, d) = \langle f_\theta(q), f_\theta(d) \rangle. \tag{1}$$

Since larger contrastive batch sizes are crucial for effective training, we implement a scalable variant of this loss. In our approach, after computing local feature representations for prefixes $\{P_1, \ldots, P_n\}$ and suffixes $\{S_1, \ldots, S_n\}$, we perform an autograd tracked `all_gather` operation on the suffixes. This operation distributes the suffix representations across all devices in the system, ensuring each device has access to all available suffixes while maintaining the graph necessary for backpropagation. As a result, each device materializes $n$ prefix embeddings and $m$ suffix embeddings (where $m$ is the total number of suffixes aggregated from all workers). The similarity scores between prefixes and suffixes form an $n \times m$ similarity matrix:

$$S_{ij} = \langle f_\theta(P_i), f_\theta(S_j) \rangle, \quad \forall i \in [1, n], \quad j \in [1, m].$$

Figure 3: **Implementation of distributed contrastive loss.** In this mock setup there are $group\_size = 4$ devices, where $group\_size \ll world\_size$. The scenario depicts the computation at Device 2, the red region shows the local features and the rest of the cells are the negatives gathered from other devices. The memory and communication cost could be controlled by $group\_size$ parameter.

However, this approach does not scale efficiently when the number of GPUs increases. Gathering representations from hundreds or thousands of GPUs incurs significant communication overhead due to the number of communicators and the size of each payload, and local memory requirements also become prohibitive as $m$ grows due to constructing a very large $n \times m$ matrix. To address this, we define distributed worker groups, where each worker gathers suffixes only from its assigned group instead of the entire system (see Figure 3). This approach constrains $m$, reducing memory overhead and allowing for efficient scaling of training across large GPU clusters.

## 2.3 BELLS AND WHISTLES

We hypothesized that the prefix-suffix retrieval pairs generated by our architecture in a self-supervised manner may include many weak or poor-quality negatives. Since each prefix is only one token away from its neighboring prefix, each suffix is similarly one token away from its neighboring suffix, and the length of each prefix has predictable relationship to the length of its corresponding suffix, the contrastive objective might overfit to syntactic patterns instead of learning meaningful semantics. To mitigate this issue, we apply a series of augmentations during training:

- **k Positive Labels**: We mark $k$ elements to the right of the main diagonal in the similarity matrix as additional positive pairs, as these neighbor prefixes may be highly similar to the true positive suffix for a given prefix, i.e. effectively "false negatives" without this adjustment (see Figure 4 $(b)$ example).
- **Lower-Diagonal Masking**: We mask out $n$ lower diagonals in the similarity matrix, excluding them from the loss calculation. Elements to the immediate left of the diagonal partially overlap with the prefix itself, constituting another type of "false negative" supervision due to their relationship to the prefix (see Figure 4 $(c)$ example).
- **Document Truncation**: We randomly truncate each of our pretraining documents such that those beyond the context length (eg. 2048) are randomly shortened such that they have final length $2048 - |\mathcal{N}(0, 100)|$, before being split into prefixes and suffixes. This reduces the chance that the model overfits to a "shortcut" feature based on the fact that the total document length $|P_i|$ minus the length of $|P_i^j|$, determines the length of matching suffix $|S_i^j|$.
- **Sparsity** We hypothesized that gathering from a large pool of negatives but then randomly subsampling them before computing the contrastive loss would enhance the diversity and therefore quality of our negatives for reasons similar to those discussed above. This sparsity inducing gather and

drop method also allows us to dial in the space-time complexity of our training objective based on model size and compute capabilities.

In the training runs for our main PSLM models, we run a supervision configuration where we combine all of these modifications to maximize the effectiveness of our unsupervised pretraining phase. See Figure 4 for illustrations of the various loss augmentations and Appendix E for an extensive ablation of each of these design choices.

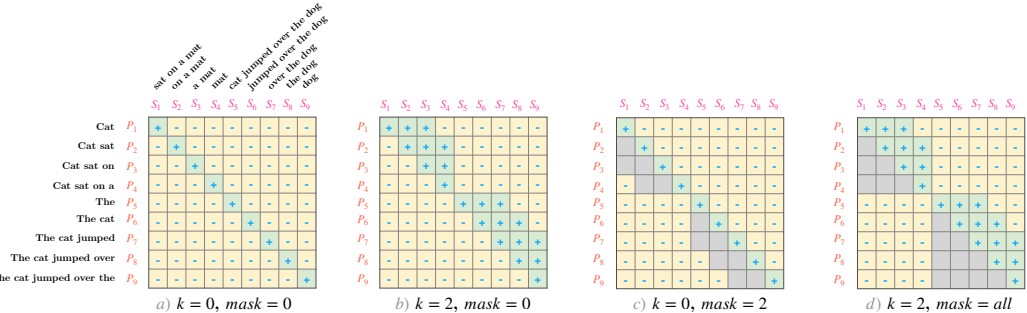

Figure 4: **Example of prefix-suffix embeddings, loss augmentations.** (a) $k = 0$ positive labels and $mask = 0$ lower diagonals is the default scenario of contrastive loss where the diagonal elements are considered positive. (b) Shows the effect of using $k = 2$ positive labels. (c) Shows when $mask = 2$ lower diagonals is used. (d) Lastly shows the compound effect on the similarity matrix when both of these augmentations are used. $mask = all$ means masking all the lower diagonal negatives. Notice that we don't cross the sequence boundary with either of these augmentations since the suffixes beyond the boundary come from different documents and thus are valid negatives for those prefixes.

## 2.4 MULTI-PHASE TRAINING

**Phase 1: unsupervised contrastive pretraining**    Initializing the model with random weights, we train using the AdamW optimizer with 6,000 warmup steps, ramping up to a peak learning rate of $2e-3$, followed by cosine decay to a minimum of $2e-4$ (for the 1.4B model we peak at $6e-04$ and decay to $6e-05$). We use a global batch size of 2048 and micro batch size of 2. We train the models for one epoch. For our production run on the WIR data mix, training lasts 120K steps ($\sim 400$B tokens). In the FineWeb-100B experiment, we train PSLM-160M for 72K steps ($\sim 100$B tokens). We set max sequence length to 2048.

For our distributed contrastive loss, we set group_size to 1,024, meaning the model gathers negatives from 1,024 devices (for the 1.4B model we gather from only 256 devices). We then randomly sample 368k negatives from this larger pool of gathered negatives to increase diversity. We also use $k = 5$ positive labels and $mask = all$ lower diagonals. We also randomly truncate each of our pretraining documents such that those beyond the context length (eg. 2048) are randomly shortened such that they have final length $2048 - |\mathcal{N}(0, 100)|$. This means that long sequences that would have been all clamped to the context length, are randomly shortened a bit more, introducing variability in sequence lengths during training.

**Phase 2: supervised contrastive training w/ pooling, positives-only**    For Phase 2 and Phase 3, we follow the standard finetuning approach in the retrieval community, specifically adopting the datasets and methodology from Nussbaum et al. (2024).

In Phase 2, we initialize the model for supervised contrastive training using the pretrained PSLM weights from Phase 1. We set a batch size of 16,384, ensuring a large number of in-batch negatives. Similar to Phase 1, we use our distributed contrastive loss implementation and a maximum sequence length of 512 to accommodate this batch size (for the 1.4B model we use a batch size of 10240 and gather negatives from 512 devices). We train using the AdamW optimizer with $\beta_1 = 0.9$, $\beta_2 = 0.95$, $\epsilon = 1e-8$, and a weight decay of 0.1. The learning rate follows a cosine decay schedule, warming up for 700 steps to a peak of $2e-5$, then decaying to a minimum of $2e-6$ (for the 1.4B model we peak

at $3e-5$ and decay to $3e-6$). We train the model on the "nomic-embed-unsupervised-data" dataset, which contains $\sim 238$M query-document pairs, for one epoch.[3]

During initial Phase 2 training experiments, we observed periodic instabilities in the training objective when naively training on all samples from each of the task-specific subsplits in order. To address this, we implement world batch "lockstep" sampling, which ensures that our global batch is filled with samples from the same single task source whilst still allowing us to sample from a different, random domain at each optimization step. This not only stabilizes training but, as in batch negatives always come from the same domain as the positives, while not explicitly mined to be difficult, the resultant negatives are trickier to differentiate than samples from other random domains would be.

**Phase 3: supervised contrastive finetuning w/ pooling, mined hard-negatives** In Phase 3, we initialize the model with PSLM weights from Phase 2 and train on the "nomic-embed-supervised-data" (Nussbaum et al., 2024), which contains $\sim 1.6$M query-document-negative triplets. We adapt the paired contrastive loss to include hard negatives in each batch. Training runs for three epochs, using 7 hard negatives per pair and a batch size of 2,048, resulting in 4,098 in-batch negatives via distributed contrastive loss (for the 1.4B model we use a batch size of 1024 consider up to 7 hard negatives and gather additional negatives from 512 devices). As in Phase 2, we employ world batch lockstep sampling. The optimizer and learning rate schedule remain unchanged from Phase 2 except we use 400 warmup steps.

## 2.5 DATASET DESIGN

We curate a data mixture comprising web, code, and instruction data that for the purposes of the work we refer to as Fine**W**eb+**I**nstruction+**R**etrieval Mix (**WIR**). The overall mixture contains approximately 400B tokens composed of $46.92\%$ diverse webtext, $22.25\%$ auxillary code-only data, $1.50\%$ high quality academic articles, $7.22\%$ instruction following examples, and $19.45\%$ retrieval tasks. For the instruction and retrieval subsets, we prepare inputs and responses or queries and documents as "web documents" via concatenation. We provide details on the subsets of each of the different types of data in Table 5. To ensure both the prefix and suffix model are equipped with the ability to handle system prompts and tasks, we construct a symmetric chat template with the system prompt inserted on the left of the query and again on the right of the document or response. In Appendix C we describe the slightly peculiar template used for our WIR mixture in full detail.

**Packing and Length distributions** For our data mixture, we pack all data sources that are not from our coding or web source splits enforcing that only sequences from the same source are packed together. We pack to a minimum of 1024 tokens or until the next example would exceed our context length. This naturally causes a each sequence in a batch to have the different lengths which turns out to be advantageous (Figure 11). The resulting distribution of sequences has an average length of 1475.04 and median length of 1400.79.

## 3 EXPERIMENTS

**Training and Validation Dynamics** We train our PSLM models following the three phase process described in Section 2.4. To efficiently iterate on model improvements, we selected three MTEB tasks—ArguAna, NFCorpus, and SciFact—as our validation datasets, chosen for their fast evaluation times. We report the progression of our validation performance in Figure 5 (Training loss and accuracy are shown in Figure 6). Once trained to the extent accommodated by our compute and dataset, we evaluate our models on the full MTEB benchmark as the test suite. Additionally, we explored generalization to instruction-following tasks, including IFEval, to assess broader applicability.

**Testing on MTEB** Recent advances in pretrained language model-based retrievers necessitate broad general-domain, zero-shot retrieval benchmarking. The Massive Text Embedding Benchmark (MTEB) (Muennighoff et al., 2023) further extends a similar previous generation collection, BEIR

---

[3]This dataset comprises paired queries and documents and so is in reality "weakly supervised" in the context of our proposed training phase breakdown, but we use the nomenclature from the original release of the dataset to be clear what we are referring to.

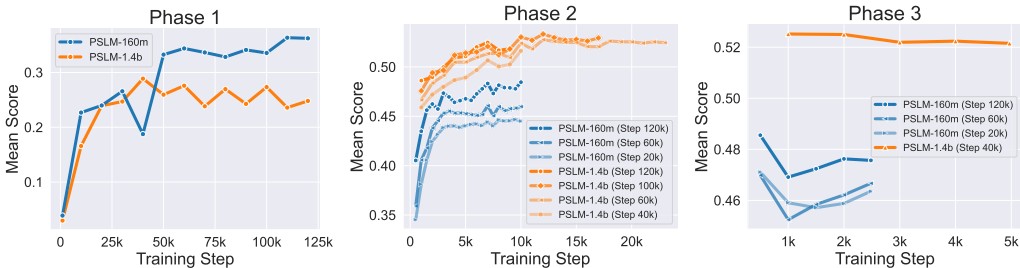

Figure 5: **Validation task performance dynamics across all three training phases.** Both of our models make steady improvements on validation tasks throughout Phase 1 and Phase 2 of training, but additional improvements during Phase 3 finetuning are less pronounced.

(Thakur et al., 2021) by adding other embedding-related non-retrieval tasks like clustering and classification. Our complete results on this diverse set of evaluations are presented in Table 1. We find that the PSLM models outperform MLM and CLM models trained under identical conditions on most subcategories of the suite. Pretraining for longer on a fortified data mixture, as well as incorporating phases of additional training on paired data also yields consistent improvements.

Table 1: **Full breakdown of performance on the MTEB suite.** It includes models with unsupervised pretraining only ("Phase 1"), finetuning on weakly supervised pairs ("Phase 2"), and finetuning on pairs with hard negatives ("Phase 3"). Results were reproduced independently unless marked with *. Model marked with $\dagger$ was initialized from a previously pretrained masked language model before unsupervised contrastive training and thus its results are marked by underline. We reserve **bold** for highlighting the top models trained using a single objective on webtext without instruction data mix-ins (despite higher absolute results for our model with enriched training data).

| Model | Avg. | Retrieval | Rerank | Cluster. | PairClass. | Class. | STS | Summ. |
|---|---|---|---|---|---|---|---|---|
| Num. Datasets -> | 56 | 15 | 4 | 11 | 3 | 12 | 10 | 1 |
| *Reference Baselines* | | | | | | | | |
| bert-base-uncased* MLM | 38.33 | 10.59 | 43.44 | 30.12 | 56.33 | **61.66** | 54.36 | **29.82** |
| contriever$^\dagger$ MLM | 50.29 | 28.76 | 51.07 | 38.24 | 77.38 | 64.93 | 71.90 | 29.87 |
| nomic-bert MLM | 26.31 | 5.84 | 34.12 | 18.54 | 33.04 | 48.61 | 33.41 | 29.04 |
| pythia-160m CLM | 35.50 | 8.23 | 40.82 | 29.60 | 48.20 | 57.40 | 51.30 | 29.48 |
| *Trained on Fineweb-100B* | | | | | | | | |
| MLM-160m | 33.44 | 7.09 | 38.72 | 22.90 | 52.37 | 54.30 | 52.30 | 28.05 |
| CLM-160m | 37.74 | 8.95 | 41.28 | 27.82 | **60.82** | 60.31 | 57.36 | 28.26 |
| PSLM-160m | **42.12** | **15.30** | **45.65** | **38.79** | 56.14 | 59.49 | **60.98** | 27.73 |
| *Finetuned on "Phase 3" data* | | | | | | | | |
| MLM-160m | 35.02 | 19.90 | 40.06 | 17.12 | 61.13 | 44.61 | 56.57 | 30.18 |
| CLM-160m | 55.20 | 45.08 | 52.1 | 37.10 | 76.68 | 65.03 | 75.72 | 31.04 |
| PSLM-160m (mean-pooled) | 55.29 | 41.23 | 52.11 | 39.42 | 81.06 | 64.98 | 78.29 | 30.17 |
| PSLM-160m (lasttoken-pooled) | 56.37 | 41.32 | 51.79 | 40.80 | 80.44 | 69.1 | 78.01 | 30.59 |
| *Trained on WIR Data Mix ("Phase 1")* | | | | | | | | |
| PSLM-160m (@120k) | 48.45 | 26.06 | 50.73 | 42.38 | 72.92 | 61.04 | 67.34 | 28.48 |
| PSLM-1.4b (@100k) | 47.70 | 18.95 | 51.29 | 43.04 | 73.48 | 62.93 | 70.33 | 29.65 |
| *Finetuned on ("Phase 2","Phase 3")* | | | | | | | | |
| PSLM-160m (Phase 2, 1ep) | 54.50 | 41.97 | 52.64 | 39.87 | 77.65 | 61.47 | 77.22 | 30.69 |
| PSLM-160m (Phase 3, 3ep) | 58.24 | 49.89 | 53.98 | 41.21 | 81.45 | 65.21 | 78.62 | 30.84 |
| PSLM-1.4b (Phase 2, 0.5ep) | 57.05 | 45.69 | 54.49 | 42.27 | 79.74 | 63.45 | 79.44 | 31.36 |

**Instruction Following and Reasoning** Instruction following is a critical capability for retrieval models, directly influencing their ability to precisely extract relevant information based on complex user objectives. By accurately interpreting and executing instructions, retrievers can filter information more effectively and support reasoning tasks that demand granular and contextually aligned information retrieval. IFEval (Sun et al., 2024) is the second version of LLMBar (Zhou et al., 2023) designed to test models ability to discern instruction following outputs. It consists of 8 different "verifiable" instruction-following subtasks such as format (selecting responses in a specific format), keywords (including specific words), and length (adhering to length restrictions). Each query has a pool of 100 candidates where only 10 documents follow the corresponding instruction. Table 2

highlights the strong instruction following capabilities of PSLM models, demonstrating a significant performance improvement when instructions are added to the query. Notably, PSLM-160m (WIR mix) achieves over a 12% increase, underscoring its ability to leverage instructions effectively. In contrast, Contriever experiences a decline in performance when additional context from instructions are introduced. This difference emphasizes the effectiveness of our objective, showcasing PSLM's adaptability to instruction-augmented queries while revealing limitations in existing retrieval models like Contriever. Table 10 lists representative examples from tasks in IFEval.

Table 2: **Full breakdown of results (nDCG@10) on IFEval instruction and response retrieval task.** AVG means average results. Results copied directly from prior work are marked with *.

| Model | Avg. | | detectable_format | | keywords | | punctuation | | change_case | | length_constraints | | combination | | detectable_content | | startend | |
|---|---|---|---|---|---|---|---|---|---|---|---|---|---|---|---|---|---|---|
| Instruction prompted? | × | ✓ | × | ✓ | × | ✓ | × | ✓ | × | ✓ | × | ✓ | × | ✓ | × | ✓ | × | ✓ |
| BM25* | 28.28 | - | 26.81 | - | 26.67 | - | 42.69 | - | 42.85 | - | 20.91 | - | 33.11 | - | 34.10 | - | 14.55 | - |
| e5-mistral-7b-instruct* | 24.98 | 32.15 | 23.04 | 32.74 | 11.51 | 58.66 | 31.98 | 33.90 | 31.34 | 21.96 | 14.60 | 22.76 | 38.96 | 24.51 | 30.24 | 32.65 | 35.03 | 35.23 |
| NV-Embed-v1* | 22.45 | 27.68 | 21.65 | 25.74 | 22.51 | 42.90 | 28.55 | 37.48 | 27.25 | 27.58 | 17.89 | 19.91 | 17.77 | 24.17 | 31.32 | 36.96 | 17.17 | 19.75 |
| GritLM-7B* | 22.03 | 38.73 | 20.75 | 34.36 | 5.51 | **65.11** | 42.05 | 33.82 | 29.92 | 40.99 | 12.87 | 29.17 | 32.33 | 49.73 | 23.13 | 45.11 | 28.59 | 31.65 |
| gte-Qwen2-1.5B-instruct* | 21.85 | 42.76 | 18.13 | 44.52 | 14.95 | 58.09 | 32.27 | 32.59 | 28.90 | 34.94 | 16.40 | 23.86 | 24.68 | 42.86 | 25.95 | **56.47** | 28.39 | 53.10 |
| contriever ("Phase 1") | 27.99 | 19.85 | 27.79 | 24.44 | 25.25 | 22.38 | 32.17 | 8.9 | 31.63 | 19.38 | 17.54 | 17.14 | 36.6 | 30.55 | 33.61 | 20.95 | 30.84 | 8.31 |
| contriever-msmarco ("Phase 2") | 26.58 | 25.15 | 27.43 | 37.4 | 31.35 | 32.64 | 28.89 | 13.18 | 21.63 | 8.85 | 18.42 | 24.43 | 25.79 | 18.81 | 36.75 | 21.1 | 26.58 | 15.15 |
| Phase 1 Checkpoints | | | | | | | | | | | | | | | | | | |
| PSLM-160m (fineweb-100b) | 32.48 | 40.58 | **39.74** | **57.81** | 11.14 | 18.13 | 61.97 | 42.86 | 41.01 | 35.08 | 20.76 | 22.7 | 22.22 | 44.35 | **36.84** | 37.43 | 24.68 | 43.28 |
| PSLM-160m (WIR data mix) | 34.17 | **46.29** | 25.65 | 47.93 | 17.78 | 31.73 | **78.13** | **68.73** | 56.5 | 58.08 | 26.78 | **33.93** | **42.42** | 54.86 | 14.88 | 23.59 | **52.93** | **67.39** |
| PSLM-1.4b (WIR data mix) | **38.12** | 44.8 | 30.92 | 41.16 | **23.48** | 32.6 | 63.67 | 47.15 | **65.34** | **60.09** | 26.84 | 28.9 | 36.92 | **57.11** | 35.58 | 53.25 | 50.62 | 58.27 |

Table 3: **Ranking performance of Quicksort and Insertion Sort queries.** Among 7 sorting algorithms, PSLM-160m (WIR) correctly retrieves 4 of the golden documents as top 1 and 6 as top 3, while Contriever retrieves 3 as top 1 and 5 as top 3.

Query: "Please find me the implementation for **quicksort**"

Ranked Documents

| # | PSLM-160m (WIR) | Contriever |
|---|---|---|
| 1 | *def sort(arr): . . . sort_recursive. . .* **Quicksort** | *def sort(arr): . . . min_idx = s . . .* **Selection** |
| 2 | *def sort(arr): . . . min_idx = s . . .* **Selection Sort** | *def sort(arr): . . . max(arr). . .* **Counting Sort** |
| 3 | *def sort(arr): . . . max(arr). . .* **Counting Sort** | *def sort(arr): . . . heapify. . .* **Heap Sort** |
| 4 | *def sort(arr): . . . for i in range(. . .* **Insertion Sort** | *def sort(arr): . . . sort_recursive. . .* **Quicksort** |
| 5 | *def sort(arr): for i in range(. . .* **Bubble Sort** | *def sort(arr): . . . while gap > 0...* **Gap Sort** |
| 6 | *def sort(arr): . . . while gap > 0...* **Gap Sort** | *def sort(arr): for i in range(. . .* **Bubble Sort** |
| 7 | *def sort(arr): . . . heapify. . .* **Heap Sort** | *def sort(arr): . . . for i in range(. . .* **Insertion Sort** |

Query: "Please find me the implementation for **insertion sort**"

Ranked Documents

| # | PSLM-160m (WIR) | Contriever |
|---|---|---|
| 1 | *def sort(arr): . . . for i in range(. . .* **Insertion Sort** | *def sort(arr): . . . min_idx = s . . .* **Selection** |
| 2 | *def sort(arr): . . . sort_recursive. . .* **Quicksort** | *def sort(arr): . . . max(arr). . .* **Counting Sort** |
| 3 | *def sort(arr): . . . min_idx = s . . .* **Selection Sort** | *def sort(arr): . . . heapify. . .* **Heap Sort** |
| 4 | *def sort(arr): . . . max(arr). . .* **Counting Sort** | *def sort(arr): for i in range(. . .* **Bubble Sort** |
| 5 | *def sort(arr): for i in range(. . .* **Bubble Sort** | *def sort(arr): . . . while gap > 0...* **Gap Sort** |
| 6 | *def sort(arr): . . . while gap > 0...* **Gap Sort** | *def sort(arr): . . . sort_recursive. . .* **Quicksort** |
| 7 | *def sort(arr): . . . heapify. . .* **Heap Sort** | *def sort(arr): . . . for i in range(. . .* **Insertion Sort** |

**Qualitative Results** To further investigate the zero shot capabilities of our models, we curate a diverse set of queries and documents spanning multiple domains. This includes retrieving sorting algorithms and code solutions to assess the model's ability to comprehend and retrieve technical content, as well as retrieving multilingual news articles. We manually select to ensure they were not previously seen by any of the evaluated models and this approach ensures the integrity of the zero-shot evaluation across different languages and contexts. Additionally, we assess the model's reasoning capabilities by applying it to elementary-level reasoning tasks such as pattern matching. These qualitative retrieval results offer valuable insights into the model's versatility and effectiveness

in handling a range of query types and domains without the need for task-specific finetuning. Since PSLM-160m (fineweb) was trained without task instructions, we observe a drop in performance across most domains when instructions are introduced. In contrast, PSLM-160m (WIR mix) consistently benefits from prompting.

We observe that Contriever (Phase 1) consistently outperforms other models (including Nomic and Contriever Phase 2) on our curated set of queries. Therefore, we select it for a head to head comparison against our strongest model PSLM-160m (WIR data mix), and show the results of this evaluation in Table 3, Table 4, and Table 9. The only exception to this is found in the pattern matching queries where PSLM-160m (FineWeb) wins against PSLM-160m (WIR Mix) and Contriever Phase 2 beats Contriever Phase 1 in retrieving the correct color. Notably, PSLM-160m (WIR Mix) exhibits high sensitivity to prompting in this task, often performing better when no explicit prompting strategies are applied.

Table 4: **Ranking performance of Hard Leetcode Problem - Sudoku Solver Retrieval.** Our best model ranks the correct Python and C++ code snippets while Contriever ranks them as 3rd. Out of the 4 languages, PSLM-160m (WIR) correctly retrieves 4 of the golden documents as top 1, while Contriever retrieves 2 out of 4 correctly as top 1.

Query: "Please find me the Sudoku Solver solution in **Python**"

| # | Models | |
|---|---|---|
| | **PSLM-160m (WIR)** | **Contriever** |
| 1 | *def isValid(…* **Python** | *function isValid(…* **JavaScript** |
| 2 | *bool isValid(…* **C++** | *class Solution…* **Java** |
| 3 | *function isValid(…* **JavaScript** | *def isValid(…* **Python** |
| 4 | *class Solution…* **Java** | *bool isValid(…* **Java** |

Query: "Please find me the Sudoku Solver solution in **C++**"

| # | Models | |
|---|---|---|
| | **PSLM-160m (WIR)** | **Contriever** |
| 1 | *bool isValid(…* **C++** | *function isValid(…* **JavaScript** |
| 2 | *def isValid(…* **Python** | *class Solution…* **Java** |
| 3 | *class Solution…* **Java** | *bool isValid(…* **C++** |
| 4 | *function isValid(…* **JavaScript** | *def isValid(…* **Python** |

## 4 RELATED WORK

Modern retrieval approaches leverage deep neural networks and large labeled datasets to achieve strong performance (Bowman et al., 2015; Nogueira & Cho, 2019; Khattab & Zaharia, 2020). Particularly relevant to our work, Contriever (Izacard et al., 2021) builds upon the a BERT pretrained backbone using an unsupervised contrastive objective based on matching document subspans to their complements. We compare to this particluar approach in our main results. Most recently however, multi-stage training recipes and large models have dominated retrieval leaderboards, where strong pipelines first pretrain on large-scale unlabeled data using a language modeling objective and then finetune using a contrastive objective on higher-quality datasets (Wang et al., 2022; Su et al., 2022; Wang et al., 2023; Li et al., 2023b; Günther et al., 2023a;b). In this vein, many contemporary leaderboard toppers start with multi-billion parameter pretrained decoder LLMs and adapt them for retrieval tasks (Li et al., 2023a; Lee et al., 2024; Muennighoff et al., 2024). In the Appendix we discuss related methods and data assets we employ in greater detail.

## 5 LIMITATIONS AND FUTURE WORK

The results of our research are promising and suggest that the same kind of scalable, unsupervised pretraining that ushered in the modern era of generative language models is also possible for retrieval models. That said, we acknowledge that many models with higher absolute performance on benchmark suites like MTEB do exist. While our PSLM models demonstrate consistent improvement under the application of additional compute, the 1.4B model's larger parameter count doesn't yield significantly improved performance in all evaluation scenarios. Even though we explored the design space to the extent that our resources would allow, we expect that certain aspects of our pipeline and implementation remain suboptimal as they are based on existing recipes for alternate retrieval architectures. We hope that our research inspires the retrieval community to develop methods that mimic key characteristics of the modern pretraining process for generative models rather than continuing to rely so heavily on supervised data to push the current state of the art.

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

## A  APPENDIX

**Software**    We build a training pipeline based on components originally developed for LLM training at scale called LitGPT and employ a flexible strategy for hybrid data and tensor parallelism (similar to PyTorch's Fully Sharded Data Parallel) called AxoNN (Singh & Bhatele, 2022; Singh et al., 2023). All final models are trained using up to 128 nodes of AMD MI250X GPUs depending on the size of the model and phase of training. We will open source the codebase and all checkpoints upon publication.

We provide an anonymized version for review purposes only:

https://anonymous.4open.science/r/retrieval-release-F2C0.

## B  EXTENDED RELATED WORK

Research in text retrieval has evolved significantly over the past few decades, transitioning from classic statistical models to modern neural-based approaches. Early work was dominated by lexical matching methods, including the Vector Space Model (Salton et al., 1975), TF–IDF (Jones, 2021), and probabilistic approaches such as BM25 (Robertson & Zaragoza, 2009). These traditional techniques often rely on bag-of-words representations, making them susceptible to semantic mismatch issues when query and document vocabularies differ.

Modern retrieval approaches leverage deep neural networks and large labeled datasets to achieve strong performance (Bowman et al., 2015). For example, Nogueira & Cho (2019) introduced a BERT-based re-ranking model that encodes queries and documents simultaneously, yielding highly accurate relevance scores. However, this approach requires re-encoding every document for each query, which can be prohibitively expensive in large-scale settings. To address this limitation, ColBERT (Khattab & Zaharia, 2020) employs a late-interaction design: queries and documents are encoded separately, and only lightweight attention is applied at inference time, greatly improving efficiency. Meanwhile, following the bi-encoder scheme, Contriever (Izacard et al., 2021) leverages the power of raw pretraining corpus to make the model capable of various retrieval settings. To generate training pairs, they sample a random span of tokens to serve as the query and use its complement—or another randomly sampled span—as the key, thus forming positive examples for the retrieval model.

More recently, multi-stage training pipelines have gained popularity, first adapting the model on large-scale unlabeled data before finetuning on higher-quality datasets (Wang et al., 2022; Su et al., 2022; Wang et al., 2023; Li et al., 2023b; Günther et al., 2023a;b). For instance, Nussbaum et al. (2024) introduced *nomic*, a fully open-sourced model that follows a multi-stage training strategy and outperforms several closed-source models (e.g., text-embedding-ada-002 (Neelakantan et al., 2022)) across various benchmarks. Another line of work enriches the retrieval context at encoding time: Morris & Rush (2024) incorporate document neighbors into the encoding process, effectively handling out-of-distribution or tail cases by leveraging additional contextual information. Additionally, some works start with pretrained decoder models and adapt them for retrieval tasks (Li et al., 2023a; Lee et al., 2024; Muennighoff et al., 2024).

## C  EXTENDED DATA DESIGN

**Chat Template.**    We experiment with three different chat template templates for each of the data sources. For instruction datasets containing instruction and response pairs like Orca, Magpie, and GenQA, we utilize:

```
f"<QUERY> {input} <QUERY> <DOC> {response} <DOC>"
```

For nomic datasets that provide the system prompt and task description (i.e clustering), we prepare them using the following template (i.e. symmetrical around the center point where the query and document meet):

```
f"<SYSTEM_PROMPT> {system prompt} <SYSTEM_PROMPT> <Task> {task
description} <Task> <QUERY> {query} <QUERY> <DOC> {document} <DOC>
<Task> {task description} <Task> <SYSTEM_PROMPT> {system prompt}
<SYSTEM_PROMPT>
```

Table 5: **The weighting of each of the data sources used.** For FineWeb and The Stack v2, we filter out anything shorter than 1024 tokens. Additionally, we epoch three times on Retrieval data longer than 1024 tokens and epoch only once on data that is shorter 1024 tokens.

| Dataset Name | Epochs | Category |
|---|---|---|
| FineWeb Sample 350BT (>1024 tokens) | 1.0 | Web |
| RedPajama ArXiv | 1.0 | Academic |
| Dolma Wiki | 1.0 | Academic |
| GenQA | 4.0 | Instruction |
| OpenOrca | 4.0 | Instruction |
| Magpie Pro 1M | 4.0 | Instruction |
| The Stack v2 (>1024 tokens) | 0.50 | Code |
| SimpleWiki (Nomic Unsupervised >1024 tokens) | 3.0 | Retrieval |
| AG News (Nomic Unsupervised >1024 tokens) | 3.0 | Retrieval |
| AltLex (Nomic Unsupervised >1024 tokens) | 3.0 | Retrieval |
| Amazon QA (Nomic Unsupervised >1024 tokens) | 3.0 | Retrieval |
| CodeSearch (Nomic Unsupervised >1024 tokens) | 3.0 | Retrieval |
| WikiAnswers (Nomic Unsupervised >1024 tokens) | 3.0 | Retrieval |
| NPR (Nomic Unsupervised >1024 tokens) | 3.0 | Retrieval |
| S2ORC Abstract-Body Index (Nomic Unsupervised >1024 tokens) | 3.0 | Retrieval |
| SQuAD (Nomic Unsupervised >1024 tokens) | 3.0 | Retrieval |
| WikiHow (Nomic Unsupervised >1024 tokens) | 3.0 | Retrieval |
| ELI5 (Nomic Unsupervised >1024 tokens) | 3.0 | Retrieval |
| S2ORC Abstract-Citation (Nomic Unsupervised >1024 tokens) | 3.0 | Retrieval |
| StackExchange Body-Body (Nomic Unsupervised >1024 tokens) | 3.0 | Retrieval |
| StackExchange Title-Body (Nomic Unsupervised >1024 tokens) | 3.0 | Retrieval |
| StackExchange Question-Question (Nomic Unsupervised >1024 tokens) | 3.0 | Retrieval |
| Wikipedia Title-Body (Nomic Unsupervised >1024 tokens) | 3.0 | Retrieval |
| Amazon Reviews (Nomic Unsupervised >1024 tokens) | 3.0 | Retrieval |
| PAQ (Nomic Unsupervised >1024 tokens) | 3.0 | Retrieval |
| S2ORC Citation-Title (Nomic Unsupervised >1024 tokens) | 3.0 | Retrieval |
| Yahoo QA (Nomic Unsupervised >1024 tokens) | 3.0 | Retrieval |
| CC News (Nomic Unsupervised >1024 tokens) | 3.0 | Retrieval |
| Quora (Nomic Unsupervised >1024 tokens) | 3.0 | Retrieval |
| S2ORC Title-Abstract (Nomic Unsupervised >1024 tokens) | 3.0 | Retrieval |
| Yahoo Title-Answer (Nomic Unsupervised >1024 tokens) | 3.0 | Retrieval |
| CNN (Nomic Unsupervised >1024 tokens) | 3.0 | Retrieval |
| GooAQ (Nomic Unsupervised >1024 tokens) | 3.0 | Retrieval |
| Reddit (Nomic Unsupervised >1024 tokens) | 3.0 | Retrieval |
| Sentence Compression (Nomic Unsupervised >1024 tokens) | 3.0 | Retrieval |
| Yahoo Title-Question (Nomic Unsupervised >1024 tokens) | 3.0 | Retrieval |
| MSMARCO Distillation (Nomic Supervised >1024 tokens) | 3.0 | Retrieval |
| NQ CoCondenser HN Mine (Nomic Supervised >1024 tokens) | 3.0 | Retrieval |
| NLI SimCSE 50 Negs (Nomic Supervised >1024 tokens) | 3.0 | Retrieval |
| Reddit Triples (Nomic Supervised >1024 tokens) | 3.0 | Retrieval |
| Medi STS Wiki Rephrasal (Nomic Supervised >1024 tokens) | 3.0 | Retrieval |
| Medi STS StackExchange Dupe (Nomic Supervised >1024 tokens) | 3.0 | Retrieval |
| Medi STS Flickr Sampled (Nomic Supervised >1024 tokens) | 3.0 | Retrieval |
| Medi SuperNLI Sampled (Nomic Supervised >1024 tokens) | 3.0 | Retrieval |
| HotpotQA HN Mine Shuffled (Nomic Supervised >1024 tokens) | 3.0 | Retrieval |
| Fever HN Mine (Nomic Supervised >1024 tokens) | 3.0 | Retrieval |
| SimpleWiki (Nomic Unsupervised <1024 tokens) | 1.0 | Retrieval |
| AG News (Nomic Unsupervised <1024 tokens) | 1.0 | Retrieval |
| AltLex (Nomic Unsupervised <1024 tokens) | 1.0 | Retrieval |
| Amazon QA (Nomic Unsupervised <1024 tokens) | 1.0 | Retrieval |
| CodeSearch (Nomic Unsupervised <1024 tokens) | 1.0 | Retrieval |
| WikiAnswers (Nomic Unsupervised <1024 tokens) | 1.0 | Retrieval |
| NPR (Nomic Unsupervised <1024 tokens) | 1.0 | Retrieval |
| S2ORC Abstract-Body Index (Nomic Unsupervised <1024 tokens) | 1.0 | Retrieval |

For the nomic datasets that only provide a task description, we instead template them like so:

Table 6: **(Continued from Table 5) The weighting of each of the data sources used.** For FineWeb and The Stack v2, we filter out anything shorter than 1024 tokens. Additionally, we epoch three times on Retrieval data longer than 1024 tokens and epoch only once on data that is shorter 1024 tokens.

| Dataset Name | Epochs | Category |
|---|---|---|
| SQuAD (Nomic Unsupervised <1024 tokens) | 1.0 | Retrieval |
| WikiHow (Nomic Unsupervised <1024 tokens) | 1.0 | Retrieval |
| ELI5 (Nomic Unsupervised <1024 tokens) | 1.0 | Retrieval |
| S2ORC Abstract-Citation (Nomic Unsupervised <1024 tokens) | 1.0 | Retrieval |
| StackExchange Body-Body (Nomic Unsupervised <1024 tokens) | 1.0 | Retrieval |
| StackExchange Title-Body (Nomic Unsupervised <1024 tokens) | 1.0 | Retrieval |
| StackExchange Question-Question (Nomic Unsupervised <1024 tokens) | 1.0 | Retrieval |
| Wikipedia Title-Body (Nomic Unsupervised <1024 tokens) | 1.0 | Retrieval |
| Amazon Reviews (Nomic Unsupervised <1024 tokens) | 1.0 | Retrieval |
| PAQ (Nomic Unsupervised <1024 tokens) | 1.0 | Retrieval |
| S2ORC Citation-Title (Nomic Unsupervised <1024 tokens) | 1.0 | Retrieval |
| Yahoo QA (Nomic Unsupervised <1024 tokens) | 1.0 | Retrieval |
| CC News (Nomic Unsupervised <1024 tokens) | 1.0 | Retrieval |
| Quora (Nomic Unsupervised <1024 tokens) | 1.0 | Retrieval |
| S2ORC Title-Abstract (Nomic Unsupervised <1024 tokens) | 1.0 | Retrieval |
| Yahoo Title-Answer (Nomic Unsupervised <1024 tokens) | 1.0 | Retrieval |
| CNN (Nomic Unsupervised <1024 tokens) | 1.0 | Retrieval |
| GooAQ (Nomic Unsupervised <1024 tokens) | 1.0 | Retrieval |
| Reddit (Nomic Unsupervised <1024 tokens) | 1.0 | Retrieval |
| Sentence Compression (Nomic Unsupervised <1024 tokens) | 1.0 | Retrieval |
| Yahoo Title-Question (Nomic Unsupervised <1024 tokens) | 1.0 | Retrieval |
| MSMARCO Distillation (Nomic Supervised <1024 tokens) | 1.0 | Retrieval |
| NQ CoCondenser HN Mine (Nomic Supervised <1024 tokens) | 1.0 | Retrieval |
| NLI SimCSE 50 Negs (Nomic Supervised <1024 tokens) | 1.0 | Retrieval |
| Reddit Triples (Nomic Supervised <1024 tokens) | 1.0 | Retrieval |
| Medi STS Wiki Rephrasal (Nomic Supervised <1024 tokens) | 1.0 | Retrieval |
| Medi STS StackExchange Dupe (Nomic Supervised <1024 tokens) | 1.0 | Retrieval |
| Medi STS Flickr Sampled (Nomic Supervised <1024 tokens) | 1.0 | Retrieval |
| Medi SuperNLI Sampled (Nomic Supervised <1024 tokens) | 1.0 | Retrieval |
| HotpotQA HN Mine Shuffled (Nomic Supervised <1024 tokens) | 1.0 | Retrieval |
| Fever HN Mine (Nomic Supervised <1024 tokens) | 1.0 | Retrieval |

```
<Task> {task description} <Task> <QUERY> <QUERY> {query} <QUERY>
<DOC> {document} <DOC> <Task> {task description} <Task>
```

## D EXTENDED MAIN RESULTS

In Figure 6, we showcase the training loss and accuracy for PSLM-160m and PSLM-1.4b as a function of gradient steps. As training progresses, the model's loss gradually decreases while accuracy improves, reflecting effective learning and convergence. This trend is expected as the model optimizes its parameters to minimize prediction errors and better distinguish between positive and negative samples.

However, during both full-duration training runs, we observe single, recoverable instabilities which manifest as abrupt step changes in loss and accuracy. For the 1.4B parameter model, we also perform a key modification of the training objective—turning on the random length truncation augmentation—later on in its main training phase. This augmentation alters the distribution of training samples, leading to a shift in the difficulty of the contrastive problem (randomization removes a spurious feature, see Appendix E). Despite this, the model quickly stabilizes, and then continues to improve in performance at a similar rate.

Importantly, our results indicate that the model has not yet reached saturation. Given its consistent improvement throughout training, we expect that extending the training duration would lead to further gains in accuracy and overall performance.

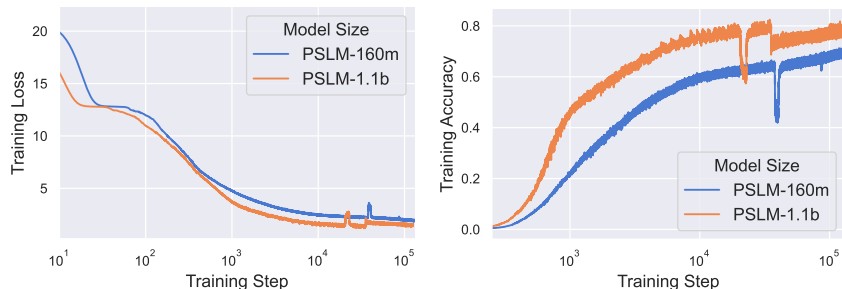

Figure 6: **Training loss and accuracy.** We observe single, recoverable training instabilities during both full-duration model runs. The step change in loss and accuracy is caused by the modification of the objective through random length truncation augmentation, which we activate later in the training of the 1.4B parameter model.

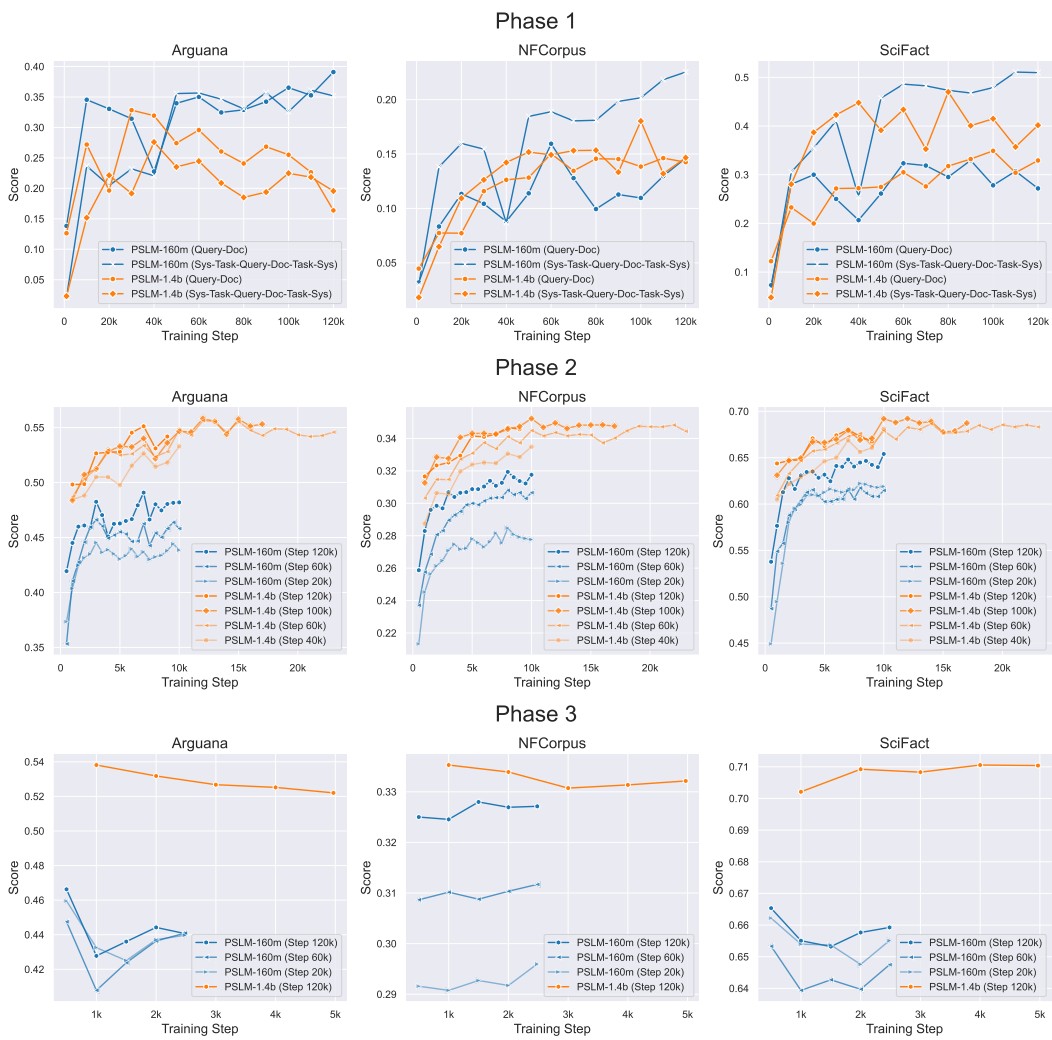

Figure 7: **Validation task performance dynamics across all three training phases.**

# E  EXPERIMENTS: ABLATIONS

**Loss augmentation**  We trained PSLM-160m on the FineWeb-100B dataset for 11 billion tokens, incorporating various combinations of $k$ positive labels and $mask$ lower diagonal augmentations. Results from our ablation studies revealed that increasing the mask on the lower diagonal consistently improved performance. Similarly, increasing $k$ initially enhanced results but began to degrade performance beyond a certain threshold (see Appendix Figure 10).

While the straightforward implementation of k positive labels would require passing a 2D probability matrix to the cross-entropy (CE) loss function—where k selected positions are marked as positive and the rest as negative—this approach forces CE to use a slower implementation instead of its optimized kernel, limiting scalability. To maintain performance, our production training randomly selects one label from the k-positive candidates to create a 1D labels tensor, ensuring compatibility with the fast CE kernel.

**Contrastive Sparsity: Negative Quantity versus Quality**  To investigate the impact of contrastive batch size, we trained PSLM on the FineWeb-100B dataset with a fixed global batch size of 256, while scaling the micro-batch size from 8 to 256 (8, 16, 32, 64, 128, 256), in other words, increasing the number of negatives. Each configuration was trained for 3.6 billion tokens using masked lower diagonals.

We observed a monotonic improvement in performance as the contrastive batch size increased, without encountering any performance plateau or decline within the tested range (see Figure 8).

Our loss augmentation strategies suggested that self-supervised retrieval pairs generated by our architecture might contain weak negatives, as most negatives (suffixes) are only one token away from their neighbors. To address this, we hypothesized that randomly sampling or dropping negatives from a larger pool-after gathering them from the distributed group-would enhance negative diversity. The results in Figure 13 confirms: for a fixed number of negatives, sampling from a larger pool improves validation performance due to increased diversity (top two labels in legend). However, reducing the number of negatives solely increases diversity and degrades performance (bottom three labels in legend).

This indicates that both quantity and quality are crucial: a large number of negatives is beneficial, but they should be sampled from a broader set to maximize diversity.

Based on these findings, our production run is trained with a large set of 368k negatives, sampled from a much larger pool: 2.95M for PSLM-160M and 737k for PSLM-1.4B, ensuring greater diversity.

**Sequence Lengths: Shortcuts and Truncation**  We observe that our objective can learn shortcuts when training sequence lengths are kept constant or when there is limited diversity in the sequence length distribution. To identify all the shortcut scenarios, we run a few experiments (see Figure 11)). First we train a model on web sequences without any intervention (called 'Baseline' in the plot), then we train a model on random tokens with random lengths, random tokens with constant length of 2048 and random tokens with the length of web sequences from the dataset. We also train with permuted batch, where we flatten the tokens of the batch and permute them. Lastly we train a model where we truncate all web sequences to 512 length. We can observe that in Figure 11 the models trained on random tokens don't converge at all. Interestingly the model trained with 512 length sequences converges faster and achieve lowest training loss and highest training accuracy. However, the validation performance suggests that the baseline performs better where we train the model on original web sequence lengths having higher length diversity. This motivated us to avoid packing training sequences to the maximum length, a common practice in LLM pretraining. Instead, we retain their original lengths. However, another issue is that if the dataset contains many sequences exceeding the maximum sequence length (2048), the length distribution becomes skewed, as these sequences get truncated to 2048. To study this effect, we conduct an ablation where we train one model using original sequence lengths without intervention and another where we truncate sequences of length 2048 to a random length between 0 and 100. We experiment with two distributions for truncation: uniform(0,100) and normal(0,100). We find that truncation improves performance on our validation tasks (see Figure 12). Based on this finding, our production training run applies normal truncation for sequences at the maximum length. However, we could not relaunch PSLM 1.4B training from scratch after this discovery and instead enabled truncation midway through its training.

**Prompting Strategies**    To study how our model performs under different prompting strategies, we run an ablation by trying out different chat templates. Figure 14 shows there is a big variance on the model's performance on validation tasks indicating the model is sensitive to certain templates. We observe that the highest validation performance achieving prompt is the one that we use during training (see the chat template that we use in Appendix C Chat Template).

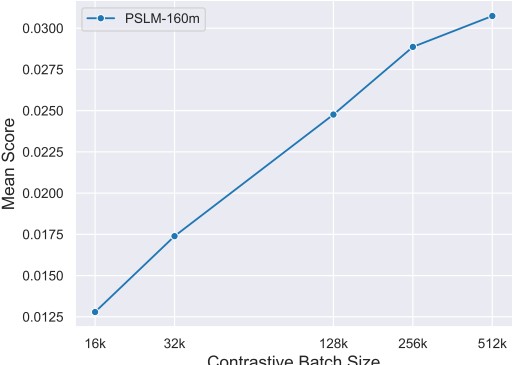

Figure 8: **Contrastive batch size scaling.** We observe monotonic increase in performance with the increase of contrastive batch size in our MTEB validation tasks.

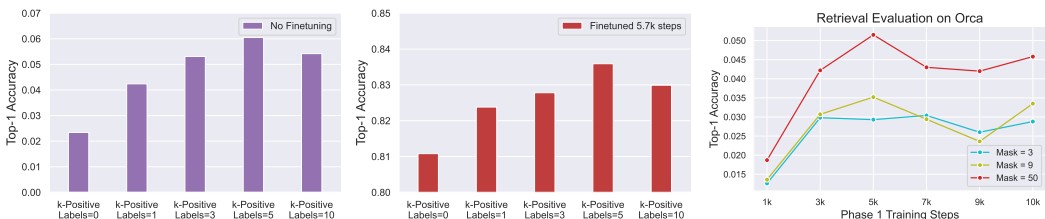

Figure 9: **Loss augmentations with k positive labels and mask lower diagonals.** (Left) We observe the effect of adding k positive labels in isolation for various $k$. After both pertaining and fine-tuning phase, higher $k$ gives better performance only until $k = 5$ and then degrades. (Right) We see the effect of mask lower diagonals in isolation and don't observe performance degradation even with the largest $mask = 50$ rather it achieves higher score. (Note: This experiment was done at very early stage of the project where we validated our model's performance by evaluating on the task of retrieving orca response from a pool of 10k and tested with 10k queries)

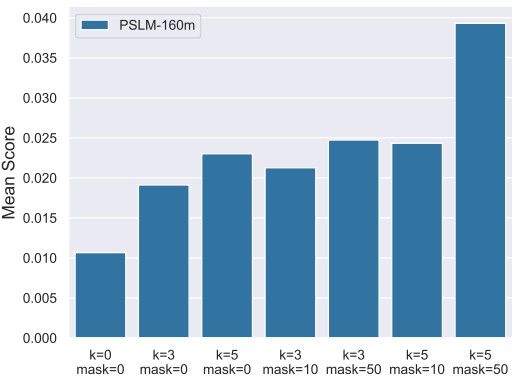

Figure 10: **Loss augmentations with masking lower diagonals and k positive labels.** We can see that training without this two augmentation don't lead to a good performance. However, with $k = 5$ and higher $mask = 50$ value, we achieve better performance. Here the mean score is done on our three MTEB validation tasks (ArguAna, NFCorpus, SciFact).

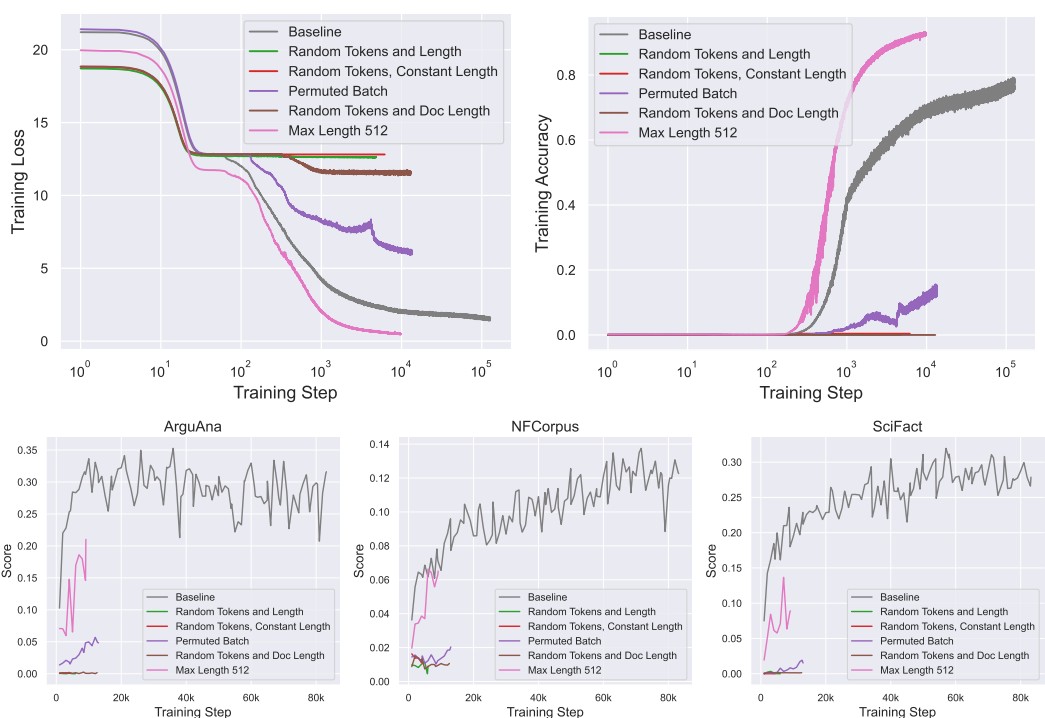

Figure 11: **Sequence length shortcut** The ablation exploration suggests that the model enjoys better validation performance when the length distribution of the sequences is more flat (Baseline), in contrast to when the length distribution is concentrated towards a single value (Max Length 512)

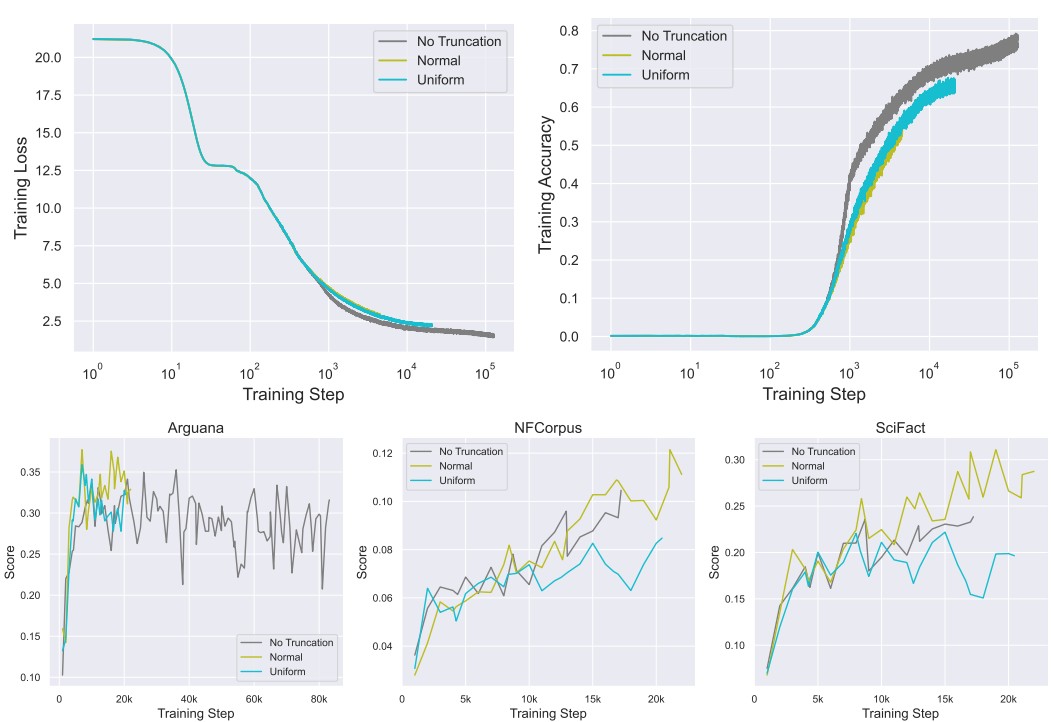

Figure 12: **Ablation on sequence truncation.**

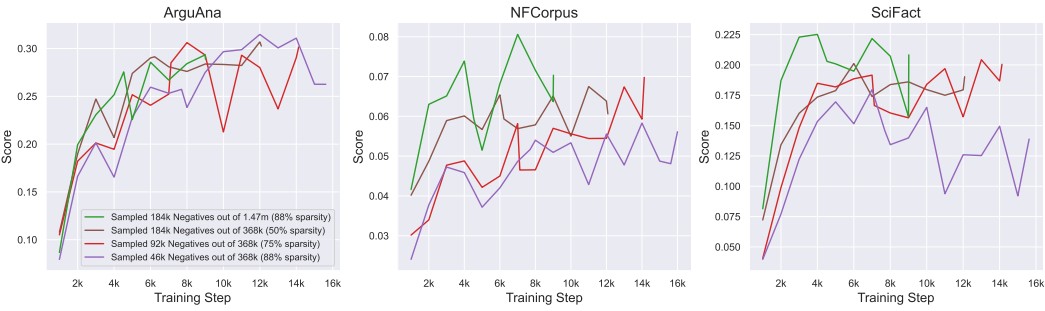

Figure 13: **Effects of gathering negative from larger pools.**

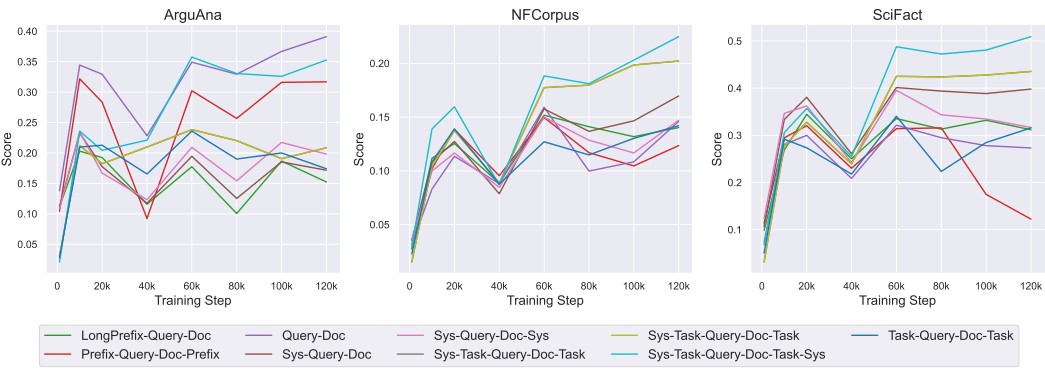

Figure 14: **Different chat templates applied to PSLM phase 1 checkpoints across training.** We can observe a high variance among the templates tested. The highest performing template (Sys-Task-Query-Doc-Task-Sys) corresponds to the chat template that was used during training.

## F MTEB

We also show the full results on a per task basis in Table 7.

| Task Name | PSLM-160M (FineWeb) | MLM-160M | CLM-160M | Contriever | PSLM-160M (WIR) |
|---|---|---|---|---|---|
| AmazonCounterfactualClassification (en) | 62.66 | 71.76 | 74.55 | 77.16 | 60.34 |
| AmazonPolarityClassification | 75.54 | 66.94 | 67.94 | 73.81 | 85.76 |
| AmazonReviewsClassification (en) | 35.76 | 32.72 | 32.5 | 37.15 | 43.76 |
| ArguAna | 22.97 | 35.37 | 40.78 | 45.05 | 35.23 |
| ArxivClusteringP2P | 37.05 | 31.19 | 32.51 | 38.32 | 46.85 |
| ArxivClusteringS2S | 24.74 | 12.32 | 17.86 | 32.52 | 34.94 |
| AskUbuntuDupQuestions | 43.38 | 46.82 | 47.27 | 54.41 | 53.93 |
| BIOSSES | 61.63 | 58.33 | 56.77 | 83.03 | 73.01 |
| Banking77Classification | 62.66 | 62.36 | 64.82 | 74.72 | 65.75 |
| BiorxivClusteringP2P | 34.8 | 29.29 | 31.84 | 31.21 | 37.82 |
| BiorxivClusteringS2S | 28.49 | 8.24 | 17.24 | 29.41 | 31.11 |
| CQADupstackRetrieval | 2.91 | 1.33 | 1.4 | 30.37 | 18.26 |
| ClimateFEVER | 8.52 | 0.63 | 2.52 | 7.16 | 12.76 |
| DBPedia | 5.63 | 0.44 | 0.18 | 27.03 | 12.91 |
| EmotionClassification | 31.98 | 26.84 | 34.47 | 40.67 | 33.03 |
| FEVER | 33.55 | 0.34 | 0.55 | 27.22 | 36.73 |
| FiQA2018 | 8.25 | 0.23 | 1.14 | 12.41 | 13.59 |
| HotpotQA | 9.48 | 2.59 | 3.32 | 41.01 | 21.4 |
| ImdbClassification | 70.59 | 68.19 | 65.77 | 66.11 | 72.95 |
| MSMARCO | 2.59 | 0.23 | 0.18 | 19.02 | 11.21 |
| MTOPDomainClassification (en) | 82.23 | 69.78 | 81.49 | 92.64 | 79.33 |
| MTOPIntentClassification (en) | 52.91 | 54.68 | 69.46 | 64.52 | 49.52 |
| MassiveIntentClassification (en) | 57.82 | 43.75 | 57.37 | 64.65 | 59.27 |
| MassiveScenarioClassification (en) | 68.81 | 45.3 | 60.11 | 71.39 | 69.11 |
| MedrxivClusteringP2P | 32.14 | 26.98 | 28.95 | 26.04 | 36.61 |
| MedrxivClusteringS2S | 32.35 | 17.21 | 22.03 | 26.78 | 33.29 |
| MindSmallReranking | 32.2 | 26.92 | 28.36 | 31.02 | 31.32 |
| NFCorpus | 19.45 | 1.97 | 2.08 | 27.17 | 22.56 |
| NQ | 5.79 | 0.04 | 0.06 | 18.06 | 14.04 |
| QuoraRetrieval | 27.27 | 49.79 | 55.69 | 83.38 | 55.25 |
| RedditClustering | 50.64 | 19.35 | 22.45 | 51.66 | 49.39 |
| RedditClusteringP2P | 52.02 | 44.33 | 46.12 | 57.39 | 57.15 |
| SCIDOCS | 6.99 | 0.64 | 0.68 | 10.97 | 12.19 |
| SICK-R | 68.53 | 52.44 | 55.79 | 67.31 | 65.97 |
| STS12 | 50.87 | 36.83 | 53.07 | 56.91 | 58.17 |
| STS13 | 64.36 | 45.94 | 55.94 | 74.47 | 67.34 |
| STS14 | 54.68 | 49.02 | 51.42 | 66.97 | 59.56 |
| STS15 | 66.06 | 60.95 | 65.09 | 78.24 | 70.2 |
| STS16 | 70.5 | 59.63 | 60.2 | 76.24 | 75.00 |
| STS17 (en-en) | 73.01 | 61.62 | 68.9 | 81.06 | 78.56 |
| STS22 (en) | 35.78 | 52.73 | 59.29 | 62.34 | 51.28 |
| STSBenchmark | 64.38 | 45.5 | 47.1 | 72.57 | 74.31 |
| SciDocsRR | 72.48 | 49.37 | 56.81 | 75.32 | 77.87 |
| SciFact | 30.89 | 6.07 | 12.07 | 57.14 | 50.84 |
| SprintDuplicateQuestions | 59.57 | 44.77 | 63.76 | 92.64 | 86.44 |
| StackExchangeClustering | 60.38 | 23.36 | 37.69 | 60.18 | 60.41 |
| StackExchangeClusteringP2P | 30.02 | 27.85 | 29.45 | 28.15 | 31.89 |
| StackOverflowDupQuestions | 29.53 | 31.78 | 32.66 | 43.52 | 39.81 |
| SummEval | 27.73 | 28.05 | 28.26 | 29.87 | 28.48 |
| TRECCOVID | 33.28 | 6.37 | 13.38 | 18.15 | 49.43 |
| Touche2020 | 11.96 | 0.34 | 0.26 | 7.25 | 24.5 |
| ToxicConversationsClassification | 60.44 | 64.45 | 63.61 | 63.83 | 58.74 |
| TweetSentimentExtractionClassification | 52.51 | 44.78 | 51.63 | 52.45 | 54.87 |
| TwentyNewsgroupsClustering | 44.09 | 11.74 | 19.9 | 39.03 | 46.74 |
| TwitterSemEval2015 | 49.11 | 45.31 | 45.82 | 57.56 | 54.76 |
| TwitterURLCorpus | 59.73 | 67.04 | 72.87 | 82.01 | 77.63 |

Table 7: Evaluation of all MTEB task models across various models.

## G EXTENDED QUALITATIVE RESULTS

As previously discussed, we curate a diverse set of queries to analyze our model's efficacy in retrieving documents in a zero-shot setting, where documents have likely not been seen during training. In particular, we acquire news articles from China Daily, Le Monde, Gujarati, El Pais, and the Guardian all dated in late January 2025. In this task, we formulate a query in a language that is different from its top retrieval hit. For example, "Pourquoi le gouvernement colombien a-t-il accepte les conditions des Etats-Unis?" inquires about Petro Urrego's rational for accepting Trump's conditions, which

is discussed in the article published by El Pais. All models tested (including nomic, cde and the contriever models) incorrectly retrieve the French article as top 1, while our model correctly retrieves the article in Spanish. This pattern is seen across different languages, showcasing PSLM's ability to transfer knowledge acquired in one language and reason before retrieving the correct document. Tables Table 8, Table 9 and Table 4 showcase various useful cases where our model outperforms other models

Table 8: **Color pattern matching queries.** From testing the models on 7 different variations of the query, contriever phase 1 retrieves the top hit correctly 3 times while contriever phase 2 retrieves the top hit correctly 5 times. Nomic is also able to correctly retrieve the correct answer 5 out of 7 times.

Query: "What comes next in the pattern: Red, Blue, Red, Blue, ?"

| | Models | |
|---|---|---|
| **#** | **PSLM-160m (WIR)** | **Contriever** |
| 1 | *Red* | *Red* |
| 2 | *Pink* | *Blue* |
| 3 | *Blue* | *Orange* |
| 4 | *Yellow* | *Green* |
| 5 | *Black* | *Pink* |
| 6 | *Green* | *Black* |
| 7 | *Orange* | *Yellow* |

Query: "What comes next in the pattern: Blue, Red, Blue, Red, ?"

| | Models | |
|---|---|---|
| **#** | **PSLM-160m (WIR)** | **Contriever** |
| 1 | *Blue* | *Red* |
| 2 | *Pink* | *Blue* |
| 3 | *Red* | *Orange* |
| 4 | *Yellow* | *Green* |
| 5 | *Black* | *Pink* |
| 6 | *Green* | *Yellow* |
| 7 | *Orange* | *Black* |

Table 9: **Ranking performance of unseen cross lingual news queries.** Our best model consistently outperforms other models in encoding the query in its language and correctly retrieving the requested document in a different language.

Query: "Pourquoi le gouvernement colombien a-t-il accepte les conditions des Etats-Unis?"

| # | Models | |
|---|---|---|
| | **PSLM-160m (WIR)** | **Contriever** |
| 1 | *Petro cede ante Trump y aceptara deportaciones para evitar la imposicion de aranceles …* **Spanish** | *La voie etroite d' Emmanuel Macron …* **French** |
| 2 | *La voie etroite d' Emmanuel Macron …* **French** | *Petro cede ante Trump y aceptara deportaciones para evitar la imposicion de aranceles …* **Spanish** |
| 3 | 针对美方可能对华加征关税，商务部作出回应… **Chinese** | *Hindi* **Hindi** |
| 4 | *Dangerous temperatures could kill 50% more Europeans by 2100 …* **English** | 针对美方可能对华加征关税，商务部作出回应… **Chinese** |
| 5 | *hindi…* **Hindi** | *Dangerous temperatures could kill 50% more Europeans by 2100 …* **English** |

Query: "欧洲与气候变化相关的伤亡人数估计有多少?"

| # | Models | |
|---|---|---|
| | **PSLM-160m (WIR)** | **Contriever** |
| 1 | 针对美方可能对华加征关税，商务部作出回应… **Chinese** | 针对美方可能对华加征关税，商务部作出回应… **Chinese** |
| 2 | *Dangerous temperatures could kill 50% more Europeans by 2100 …* **English** | *Hindi* **Hindi** |
| 3 | *La voie etroite d' Emmanuel Macron …* **French** | *Petro cede ante Trump y aceptara deportaciones para evitar la imposicion de aranceles …* **Spanish** |
| 4 | *Hindi* **Hindi** | *La voie etroite d' Emmanuel Macron …* **French** |
| 5 | *Petro cede ante Trump y aceptara deportaciones para evitar la imposicion de aranceles …* **Spanish** | *Dangerous temperatures could kill 50% more Europeans by 2100 …* **English** |

# H IFEᴠᴀʟ

We provide some practical test examples found in IFEval. All instructions listed in Table 10 end with "Any response that does not fully comply with this instruction should be considered irrelevant."

Table 10: **Examples of Queries and Corresponding Documents for the 8 Different Tasks in IFEval.** Two examples are shown for each task where our model achieves an NDCG@1 of 1.

| Task | Instruction | Query | Document |
|---|---|---|---|
| Detectable Format | Retrieve responses to the given question. It is crucial that the responses strictly follow to the specified instruction: **Highlight at least 3 sections in your answer with markdown, i.e. *highlighted section*.**. | Write a funny post for teenagers about a restaurant called "Buena Onda" which serves Mexican food | Welcome to Buena Onda - Where Tacos Find Their True Calling! . . . **Taco Tuesday Every Day** . . . **¡Hola Happiness!** . . . |
| | Retrieve responses to the given question. It is crucial that the responses strictly follow to the specified instruction: **Your answer must contain a title, wrapped in double angular brackets, such as «poem of joy».**. | A sales pitch for moms is needed for a new kind of diaper that is designed to be more comfortable for babies. | <title> The Ultimate Must-Have Diaper for Happy Babies! . . . |
| Keywords | Retrieve responses to the given question. It is crucial that the responses strictly follow to the specified instruction: **Include keywords ['ink', 'memoirs'] in the response.**. | Please rewrite the following sentence in a serious tone, similar to the president of the United States, and without an upward bias or inflation alarmism · · · | The key aspects that warrant our attention are as follows: Firstly, it is imperative for the ECB to reconsider its projections, which currently display a no- table inclination towards an upward trajectory. Sec- ondly, the ECB should maintain independence |
| | Retrieve responses to the given question. It is crucial that the responses strictly follow to the specified instruction: **In your response, the letter t should appear less than 8 times.**. | Write a logic quiz for teens about a chesterfield. | Sure! Here is a logic quiz: Question: What has a back, but no arms? |
| Punctuation | Retrieve responses to the given question. It is crucial that the responses strictly follow to the specified instruction: **In your entire response, refrain from the use of any commas.**. | Can you give me two different formal alternatives to "What's up? I'm going to the beach today" | Certainly. One alternative could be "How are you? I am planning to visit the beach later today." Another option might be "Greetings. I'll be heading to the beach today". |
| | Retrieve responses to the given question. It is crucial that the responses strictly follow to the specified instruction: **In your entire response, refrain from the use of any commas.**. | What is the next number in this series: 2, 2, 4, 12, 48? Please use mathematical notation in your answer. | The next number in the series is 240 |
| Change Case | Retrieve responses to the given question. It is crucial that the responses strictly follow to the specified instruction: **Your entire response should be in English, and in all lowercase letters. No capital letters are allowed.**. | Write a detailed review of the movie "The Social Network" in markdown format. | The Social Network* is a captivating film that tells the story of the founding of Facebook and the legal battles that followed. Directed by . . . |
| | Retrieve responses to the given question. It is crucial that the responses strictly follow to the specified instruction: **Your entire response should be in English, and in all lowercase letters. No capital letters are allowed.**. | Write a cover letter for a job application as a tour guide in Beijing. | Dear hiring manager, I am excited to submit my application for the tour guide position in Beijing. With my strong communication skills and passion for sharing knowledge with others . . . |
| Length Constraints | Retrieve responses to the given question. It is crucial that the responses strictly follow to the specified instruction: **Your response should contain less than 17 sentences.**. | A young couple that just got married is going to Seattle for two days. They're flying from New York. Could you write them an itinerary? | Day 1:
- Arrive in Seattle and check in to your hotel
- Visit Pike Place Market for a morning filled with fresh food and local vendors
- Enjoy lunch at a seafood restaurant along the waterfront . . . |
| | Retrieve responses to the given question. It is crucial that the responses strictly follow to the specified instruction: **Your response should contain less than 7 sentences.**. | Write a template for how to calculate the offset of an element in an array. | Certainly! Here is a template with less than 7 sentences to calculate the offset of an element in an array: To calculate the offset . . . |
| Combination | Retrieve responses to the given question. It is crucial that the responses strictly follow to the specified instruction: **Give two different responses. Responses and only responses should be separated by 6 asterisk symbols: ******.**. | Create a dialogue between two people who are both trying to avoid using the letter t. | Certainly! Here is the dialogue: Person 1: "Hello, how are you feeling?" Person 2: "I'm doing well, thank you. How about yourself?" |
| | Retrieve responses to the given question. It is crucial that the responses strictly follow to the specified instruction: **Give two different responses. Responses and only responses should be separated by 6 asterisk symbols: ******.**. | A nucleus is a cluster of protons and neutrons. Elaborate on this in a list of facts. | Protons and neutrons are held together by strong nuclear force to form the nucleus. The number of protons in the nucleus determines the element's atomic number |

Table 11: **(Continued from Table 10) Examples of Queries and Corresponding Documents for the 8 Different Tasks in IFEval.** Two examples are shown for each task where our model achieves an NDCG@1 of 1.

| Task | Instruction | Query | Document |
|---|---|---|---|
| Detectable Content | Retrieve responses to the given question. It is crucial that the responses strictly follow to the specified instruction: **At the end of your response, please explicitly add a postscript starting with P.P.S**. | 11 results have an average of 51. The first five have an average of 49, and the last nine have an average of 52. What was the sixth result? | The sixth result, denoted as $x$, can be found using the concept of weighted averages. Since the eleven results have an average of 51, the sum of all results is $11 \times 51 = 561$. .... |
| | Retrieve responses to the given question. It is crucial that the responses strictly follow to the specified instruction: **At the end of your response, please explicitly add a postscript starting with P.P.S**. | List up to 10 possible names for a new baby boy. | 1. Ethan 2. Oliver 3. Benjamin ... |
| Startend | Retrieve responses to the given question. It is crucial that the responses strictly follow to the specified instruction: **Finish your response with this exact phrase Is there anything else I can help with?. No other words should follow this phrase.**. | What is multivariate analysis? Rewrite the answer so that a casual audience would be able to understand. | Multivariate analysis is a statistical technique used to ... |
| | Retrieve responses to the given question. It is crucial that the responses strictly follow to the specified instruction: **Finish your response with this exact phrase Any other questions?. No other words should follow this phrase.**. | I'm a 12th grader and I need some help with my college applications. Can you give me some advice? | - Start early
- Choose schools wisely
- Write a compelling essay ... |