# OpenReview forum: "Scalable Pretraining of Retrieval Models"
_ICLR.cc/2026/Conference — Submitted to ICLR 2026_

### Official Review · Reviewer_cSe9 · 2025-10-27

**Soundness:** 2
**Presentation:** 3
**Contribution:** 2
**Rating:** 4
**Confidence:** 4

**Summary:**

The paper proposes Prefix–Suffix Language Modeling (PSLM) as a scalable pretraining framework for retrieval models. It divides each text sequence into a prefix (treated as a “query”) and a suffix (treated as a “document” or “answer”), then uses causal and anti-causal masking to obtain two representations. A contrastive InfoNCE loss brings representations from the same sequence closer while pushing apart those from different sequences. The authors further introduce a three-stage training process: self-supervised pretraining, weakly supervised positive pairs, and optional hard-negative finetuning, and claim improved retrieval performance over MLM/CLM baselines with better scalability.

**Strengths:**

1. The paper addresses an important problem in retrieval pretraining, aiming to align the pretraining objective more closely with real-world query–document matching tasks.
2. The prefix–suffix formulation provides an intuitive way to model “query–answer” correspondence within the same sequence, keeping the architecture compact and training objective straightforward.
3. Experiments are conducted across multiple retrieval benchmarks, showing stable and consistent gains over MLM and CLM baselines, with additional ablation and qualitative analyses.

**Weaknesses:**

1. The paper’s main limitation lies in its limited conceptual novelty. The prefix–suffix contrastive pretraining objective closely resembles prior methods such as RetroMAE, CoCondenser, and Contriever. These works already explored similar contextual or sentence-level contrastive schemes, and the paper does not clearly articulate what PSLM adds beyond re-framing or scaling.
2. The contribution is mainly engineering-oriented rather than conceptual. Most of the performance gain seems to come from large-scale training, distributed optimization, and more negatives, rather than a fundamentally different modeling principle.
3. Several recent and strong retrieval or embedding models such as E5, GTR, and modern BGE variants are not included in comparison, which weakens the empirical strength of the SOTA claim.
4. The claim of scalability is insufficiently supported. The paper lacks iso-compute or iso-data comparisons, GPU-hour cost curves, or throughput analysis. Without quantitative cost–performance trade-offs, the “scalable” claim remains mostly qualitative.

**Questions:**

1. Can you provide iso-compute or iso-data comparisons with MLM/CLM and other recent retrieval pretraining methods?
2. How exactly does your scalable implementation improve training efficiency? Please include GPU-hour and throughput metrics.

---

> ### Author Response · Authors · 2025-11-20
>
> We thank the reviewer for their feedback. We believe some of these concerns stem from a misunderstanding of our core objective and its differentiation from prior art, which we will clarify below. We hope that, in light of these clarifications, the reviewer might reconsider their score.
>
> > 1. The paper’s main limitation lies in its limited conceptual novelty. The prefix–suffix contrastive pretraining objective closely resembles prior methods such as RetroMAE, CoCondenser, and Contriever.
>
> We respectfully disagree about the limited novelty and similarity to other methods. While they are contrastive objectives, the methodological difference is the core of our contribution.
>
> - RetroMAE & CoCondenser are MLM-based objectives. The entire premise of our work, as stated in the Introduction, is that MLM-style objectives are fundamentally misaligned with retrieval. Our PSLM objective is proposed as a replacement for this paradigm, not an increment of it.
> - Contriever uses a span-complement objective (a random span vs. the rest of the document). Our prefix-suffix objective is fundamentally different: it is a sequential, causal-like split that models retrieval as a "document completion" task. We argue this is a more natural analog to the query-document task and is more aligned with the generative pretraining paradigm than Contriever's random-span objective.
>
> > 2. The contribution is mainly engineering-oriented rather than conceptual. Most of the performance gain seems to come from large-scale training, distributed optimization, and more negatives…
>
> We argue that for our method, the conceptual (PSLM) and engineering (Sec 2.2, 2.3) contributions are inseparable. The conceptual PSLM objective cannot be run naively at scale; it requires the distributed contrastive loss and loss augmentations we developed. These "engineering" contributions are precisely what enable the conceptual objective to be "scalable," allowing it to run on web-scale data and benefit from large negative batches.
>
> > 3. Several recent and strong retrieval or embedding models such as E5, GTR, and modern BGE variants are not included in comparison…
>
> This is a deliberate choice of scope. Our paper is an "apples-to-apples" comparison of unsupervised pretraining objectives (PSLM vs. MLM vs. CLM) trained from scratch on the same data.
>
> Models like E5 and BGE are closer to "tech reports with open-weights". Their training relies heavily on massive, curated paired datasets, which we classify as "supervised" (Phase 2/3) data. They are not valid baselines for our Phase 1 (unsupervised) contribution. Our goal is not to beat the SOTA, but to demonstrate that a better pretraining objective can yield strong zero-shot retrieval capabilities without this reliance on expensive, supervised paired data, which is a key bottleneck in specialized domains such as legal, medical, finance, code where you may only have a bunch of text documents at your disposal without any supervised retrieval data.
>
> > 4. The claim of scalability is insufficiently supported. The paper lacks iso-compute or iso-data comparisons, GPU-hour cost curves, or throughput analysis.
> 5. Can you provide iso-compute or iso-data comparisons with MLM/CLM and other recent retrieval pretraining methods?
> 6. How exactly does your scalable implementation improve training efficiency? Please include GPU-hour and throughput metrics.
>
> This is a fair point. Our "scalable" claim refers to the design (i.e., being fully self-supervised from raw web text) rather than a quantitative benchmark of GPU-hours, which was beyond our "one-time consumable compute budget". Our resources were invested in answering the scientific question, not in engineering-cost benchmarking.
>
> However, to answer Q1, we do provide a direct iso-data comparison in Figure 2. This figure compares PSLM-160m, MLM-160m, and CLM-160m, all trained on the same 100B token dataset. This is the core "apples-to-apples" comparison that demonstrates the superiority of our pretraining objective in the zero-shot setting.
>
> To answer Q2, we did not benchmark raw throughput. The novelty of our scalable implementation (Sec 2.2) is not in raw FLOPS, but in its ability to control the prohibitive cost of a massive contrastive loss. By gathering negatives only within a group_size, we provide a mechanism to trade off negative batch size (and performance) with communication/memory cost, allowing the objective to run on large GPU clusters.

---

### Official Review · Reviewer_rubc · 2025-10-29

**Soundness:** 2
**Presentation:** 3
**Contribution:** 2
**Rating:** 2
**Confidence:** 4

**Summary:**

The paper proposes Prefix-Suffix Language Modeling (PSLM), a self-supervised contrastive pretraining approach for retrieval models that aims to parallel next-token prediction in generative LMs. Instead of MLM/CLM objectives, the model learns to align prefix embeddings from a causal transformer with suffix embeddings from an anti-causal transformer at every token position. The authors introduce several training “bells and whistles” (e.g., distributed contrastive loss, multi-positive labels, diagonal masking) and demonstrate that PSLM scales to web-scale corpora and improves zero-shot retrieval and instruction-following ability on benchmarks such as MTEB and IFEval. They further show significant improvements when combined with supervised phases.

**Strengths:**

1. An important topic is studied, scalable pre-training for dense retrieval.
2. The analogy of the proposed PSLM to next-token prediction is interesting.
3. The distributed contrastive loss is useful for large-scale retrieval training.
4. The MTEB and IFEval benchmarks are used for evaluation, each including a wide range of datasets.

**Weaknesses:**

1. The novelty of PSLM is limited and the relationship to prior work is under-positioned. There is a large body of works on pre-training for information retrieval, where the pre-training objectives are not limited to masked language modeling and next token prediction. PSLM is very close to contrastive pretraining using prefix-complement spans, as used in Contriever and Contrastive Predictive Coding-style objectives (Oord et al., 2018). The conceptual novelty appears incremental, though framed as foundational. Citation discussion acknowledges Contriever yet undersells methodological similarity.

Other pre-training objectives, such as "RetroMAE: Pre-Training Retrieval-oriented Language Models Via Masked Auto-Encoder", although also based on MLM, are also very powerful. They are not discussed or compared in the paper.

2. The method descriptions have discrepancy in Figure 1 and Figure 4. Figure 1 shows that the first token (suffix) is paired with the last token (prefix), while in Figure 4, the first token (suffix) is paired with the first token (prefix). This is a bit confusing. If Figure 1 describes the method accurately, matching the first token and the last token does not really make sense to me.

3. The performance of stronger models on MTEB is not reported, such as RetroMAE. Baselines are not sufficient. Nomic-Embed also has better performance (59.9) than reported in Table 1.

4. Related work based on large language models are not mentioned:
Llm2vec: Large language models are secretly powerful text encoders
Finetuning llama for multi-stage text retrieval.
Llama2Vec:Unsupervised Adaptation of Large Language Models for Dense Retrieval
Unleashing the Power of LLMs in Dense Retrieval with Query Likelihood Modeling

5. The citation of dataset IFEval (Sun et al., 2024) is wrong. Sun et al., 2024 introduce "MAIR: A Massive Benchmark for Evaluating Instructed Retrieval" for retrieval tasks, but IFEval is from paper "Instruction-Following Evaluation for Large Language Models" and constructed for LLM evaluation. Compared to IFEval, MAIR is better for evaluation.

6. There is potential data leakage. Some datasets used in validation are likely included in the retrieval portion of WIR data. The paper should explicitly quantify data contamination risk.

**Questions:**

Please see above.

---

> ### Author Response · Authors · 2025-11-20
>
> We thank the reviewer for their detailed feedback and for identifying several areas for clarification and improvement. We will respond to each of the specific comments made in the review. We hope that in light of these clarifications, the reviewer might reconsider their score.
>
> > 1. The novelty of PSLM is limited and the relationship to prior work is under-positioned. PSLM is very close to Contriever... Other objectives, such as "RetroMAE"... are not discussed or compared.
>
> We respectfully disagree about the limited novelty and similarity to Contriever. While both are contrastive objectives, the methodological difference is the core of our contribution.
> - Contriever uses a span-complement objective: a random span of tokens is used as the "query," and its complement (the rest of the document) is the "positive."
> - PSLM uses a prefix-suffix objective: the document is split sequentially, where text[0:i] (the prefix) is the "query" and text[i:n] (the suffix) is the "positive."
>
> This sequential formulation encourages the model to align early-document representations with the larger continuation of the document, yielding a retrieval signal that is more stable and semantically coherent than a random-span pairing. This distinction matters. The random-span objective mixes unrelated spans and can produce positives with weaker semantic coupling, whereas our prefix-suffix formulation systematically captures the natural progression of a document. We argue this provides a more direct and retrieval-aligned training signal, particularly in settings involving long-form or generative models, as detailed in our motivation.
>
> Regarding RetroMAE, we thank the reviewer for the reference. We will happily add RetroMAE to our related work as a strong example of the MLM-based paradigm we are providing an alternative to.
>
> > 2. The method descriptions have discrepancy in Figure 1 and Figure 4. Figure 1 shows that the first token (suffix) is paired with the last token (prefix), while in Figure 4, the first token (suffix) is paired with the first token (prefix).
>
> We apologize for the confusion; this is a misunderstanding of the diagrams, which we will clarify.
> - Figure 1 is a high-level schematic. It is a birds-eye view of the architecture showing how prefixes and suffixes are created using “causal” and “anti-causal” attention. It shows prefix features ($P_i$) and suffix features ($S_j$) being compared. The green checkmarks on the diagonal indicate that the model is trained to match $P_1$ with $S_1$, $P_2$ with $S_2$, etc. It does not show a "first-to-last" token match.
> - Figure 4 is a concrete example of this diagonal matching, showing what happens inside the objective function, exactly which prefixes and suffixes we’re matching. The first row (prefix "Cat") is the $i=1$ prefix. The first column (suffix "sat on a mat...") is the $j=1$ suffix. The cell (1,1) is the positive pair. The second row (prefix "Cat sat") is the $i=2$ prefix, which is matched with the second column (suffix "on a mat."), and so on.
>
> We will better clarify the figures in our writing of the camera-ready version of the paper.
>
> > 3. The performance of stronger models on MTEB is not reported, such as RetroMAE. Baselines are not sufficient. Nomic-Embed also has better performance (59.9) than reported in Table 1.
>
> Our goal in this paper is not to achieve a new SOTA on MTEB, but to perform a rigorous, "apples-to-apples" comparison of pretraining objectives (PSLM vs. MLM vs. CLM) trained from scratch on the same data.
>
> The reviewer is correct that the official nomic-embed model achieves a high score like 59.9. This score is for their model trained with their full multi-phase pipeline and various sources of supervised data mix. The nomic-bert MLM in our table 1 is a reference bert model we use to compare with our own MLM baseline. This is a crucial distinction. We will add RetroMAE to our related work, but as per point 1, it represents a different line of research (MLM-based) than our novel objective.
>
> > 4. Related work based on large language models are not mentioned…
>
> We thank the reviewer for these excellent and relevant references. We will add a discussion them in the camera-ready version.
>
> > 5. The citation of dataset IFEval (Sun et al., 2024) is wrong... MAIR is better for evaluation.
>
> We specifically use MAIR’s IFEval task for our evaluation (not the eval code designed for LLMs), which is why we cite MAIR. However, for better reference to the original eval, we will also include a citation of the original work.
>
> > 6. There is potential data leakage. Some datasets used in validation are likely included in the retrieval portion of WIR data.
>
> Our WIR data mix uses the Nomic unsupervised and supervised training data splits, which are explicitly detailed in Table 5 and Table 6. Moreover, to show the raw capability of our objective, we do standalone training on Fineweb from scratch to compare with the baselines.

---

### Official Review · Reviewer_8aH2 · 2025-10-31

**Soundness:** 2
**Presentation:** 3
**Contribution:** 2
**Rating:** 4
**Confidence:** 3

**Summary:**

This work introduces a novel approach for self-supervised training of a retriever. The core idea involves selecting the correct suffix (i+1, L) from a candidate set using the representation of a prefix (0, i), where the prefix and suffix representations are obtained through left-to-right and right-to-left Transformers, respectively. This method enables large-scale unsupervised training of retrieval models and achieves improved performance on the MTEB benchmark.

**Strengths:**

1. The approach of predicting the suffix using the prefix demonstrates a certain degree of novelty.
2. Experimental results indicate a positive improvement, although it almost levels off after supervised fine-tuning (SFT).

**Weaknesses:**

1. There is a lack of sufficient discussion and insight into why this method performs better than next token prediction during pre-training. Both methods essentially involve predicting future text based on a prefix—one predicts the very next token, while the other predicts the corresponding suffix. However, suffix prediction inherently requires roughly double the computational cost. A deeper exploration of why pre-training via suffix prediction yields superior results would be highly valuable.
2. Table 1 shows that PSLM-160M exhibits almost no improvement over CLM-160M after Phase 3 fine-tuning. This observation further reinforces the concerns raised in my first point.
3. In Table 1, the performance of PSLM-1.4B is worse than that of PSLM-160M. This raises concerns about whether the performance gain of this method scales reliably with increasing model parameters.
4. The discussion of related work could be strengthened. Several other works have also explored scalable training for retrieval, fully within an autoregressive loss formulation, such as https://arxiv.org/abs/2306.13421 and https://arxiv.org/abs/2410.01651.

**Questions:**

See weaknesses.

---

> ### Author Response · Authors · 2025-11-19
>
> We thank the reviewer for their thoughtful feedback. We believe there is a crucial clarification to be made about our objective (point 1) that will resolve the reviewer's other concerns (points 2 and 3). We hope that in light of these clarifications, the reviewer might reconsider their score.
>
> > 1. There is a lack of sufficient discussion and insight into why this method performs better than next token prediction... Both methods essentially involve predicting future text based on a prefix... suffix prediction inherently requires roughly double the computational cost.
>
> This is the most critical point, and we apologize if our paper was not clear. The PSLM objective is fundamentally different from a next-token prediction (CLM) objective.
> - A **Causal Language Model (CLM)** performs a classification task. It predicts the single next token from a fixed-size vocabulary (e.g., 32,000 tokens). The specific loss function is the cross entropy between the model logits and the 1-hot label representing the ground truth next token.
> - Our **Prefix-Suffix Language Model (PSLM)** performs a retrieval task. It uses a prefix embedding to find its corresponding entire suffix embedding from a massive, dynamic pool of negative suffix embeddings (e.g., 368k+ negatives gathered from other documents ). The specific loss function is the InfoNCE loss where a (single) prefix embedding and a set of suffix embeddings are turned into a similarity vector, and the goal is to make the similarity between the prefix and it’s matching suffix high, while making the other similarities low, which is represented as minimizing the cross entropy between the similarity score vector (logits) and the 1-hot label representing the correct suffix this prefix should match to.
>
> In summary, PSLM does not predict tokens. Instead, it learns to match a query-like prefix to its document-like suffix via a contrastive loss. A causal and anti-causal pass is required to create these prefix/suffix representations, which are then used in a task that is directly aligned with retrieval. We argue this is a more suitable pretraining objective for a retrieval model than simple next-token classification.
>
> > 2. Table 1 shows that PSLM-160M exhibits almost no improvement over CLM-160M after Phase 3 fine-tuning. This observation further reinforces the concerns raised in my first point.
>
> This is an insightful observation that gets to the core of our paper's motivation. Our primary claim is about the unsupervised pretraining stage (Phase 1).
>
> As shown in Figure 2, in the zero-shot setting (after Phase 1), our PSLM objective vastly outperforms the CLM baseline. This demonstrates the "natural affinity" of our pretraining objective for retrieval.
>
> The fact that the gap narrows after massive supervised fine-tuning (Phases 2 and 3) is interesting. It suggests that this specific fine-tuning recipe is powerful enough to lift a CLM baseline to a strong performance level. However, our paper's goal was to design a pretraining objective that provides strong zero-shot retrieval capabilities without relying on this expensive supervised fine-tuning, which is a key bottleneck in specialized domains such as legal, medical, finance, and code, where you may only have a bunch of text documents at your disposal without any supervised retrieval data.
>
> > 3. In Table 1, the performance of PSLM-1.4B is worse than that of PSLM-160M. This raises concerns about whether the performance gain of this method scales reliably with increasing model parameters.
>
> Regarding the lack of a large gap between the performance of the 160m and 1.4b model, we agree that the results are slightly surprising. That said, going from 160m parameters to above 1B parameters (even for standard LLM training) represents a large shift in the parallelization factor and compute node counts required for a single job, and so we were only able to do the more extensive ablations of the various design choices described in Sec 2.2 and 2.3 at the smaller 160m scale. As a result, we expect that our design hyperparameters were simply suboptimal for the larger 1.4B model, where we could not search for optimal hyperparameters exhaustively. We remark that the “sensible defaults” we all have grown accustomed to for normal Llama-like LLM pretraining that allow the easy scaling of standard experiments without grid search just don’t exist in our novel setting.
>
> > 4. The discussion of related work could be strengthened. Several other works have also explored scalable training for retrieval...
>
> We thank the reviewer for these valuable references. They are indeed relevant, and we will gladly incorporate a discussion of these contemporary works into the related work section in the camera-ready copy.

---

### Official Review · Reviewer_PCeL · 2025-10-31

**Soundness:** 3
**Presentation:** 2
**Contribution:** 3
**Rating:** 6
**Confidence:** 3

**Summary:**

This paper proposes prefix-suffix language modeling (PSLM), a novel approach pertaining to retrieval models. Learned lessons from pretraining generative models, PSLM aligns the prefix and the suffix in the embedding space to improve the retrieval performance. This paper also addresses scalability issues when using contrastive learning in distributed scenarios and introduces several pre-training tricks. This paper pretrains the retrieval models in a multi-phase training scheme. Experiments show that with PSLM, the pretrained models can produce state-of-the-art performance compared to traditional pre-training schemes.

**Strengths:**

This paper identified the pretraining issue of retrieval models and addressed it using a proper method.

**Weaknesses:**

1. It would be better to provide additional results across a wider range of model sizes, such as 320M, 640M, 3B, and 7B, to show the scalability of the proposed methods.
2. The paper writing is quite poor. For example, the paper lacks an explanation of the design philosophy and only introduces the specific steps of their methods.

**Questions:**

1. Can we merge phase 2 with phase 3 into one phase?
2. How about the performance of the proposed method in the long-context scenarios?

---

> ### Author Response · Authors · 2025-11-19
>
> We thank the reviewer for their feedback on our work. We will respond to each of the specific comments made in the review in hopes of at least providing additional clarity as to certain choices made throughout the research and writing process. We hope that if any of the reviewer’s concerns are adequately addressed that they might reconsider their score in the direction of acceptance.
>
> > 1. It would be better to provide additional results across a wider range of model sizes, such as 320M, 640M, 3B, and 7B, to show the scalability of the proposed methods.
>
> We agree with the reviewer that a broader study of model sizes would be valuable. Our experiments were conducted using a one-time, limited computational grant, as we detail in the paper. Scaling PSLM requires ablating novel hyperparameters (e.g., group_size for the distributed loss , k positive labels , and masking ), which we were only able to perform at the 160M scale. These hyperparameters are likely suboptimal for the 1.4B model, which we believe explains the smaller-than-expected performance gap.
>
> Given this, running new ablations for a wide range of new model sizes (320M, 640M, 3B, 7B) was computationally infeasible. We chose to focus our limited budget on a thorough "apples-to-apples" comparison against MLM and CLM baselines at the 160M scale and one larger 1.4B model to demonstrate the viability of the objective at scale. We believe these results are sufficient to show the promise of our pretraining method, and we leave a more exhaustive scaling-law analysis to future work.
>
> > 2. The paper writing is quite poor. For example, the paper lacks an explanation of the design philosophy and only introduces the specific steps of their methods.
>
> We are sorry the reviewer felt the writing was poor, and we will take this feedback into account for the camera-ready version.
>
> Regarding the "design philosophy," we believe this is the central thesis of our Introduction (Section 1) and the Motivation for the PSLM objective (Section 2). To be explicit, our philosophy is:
> - **The Problem:** The dominant MLM/CLM pretraining tasks are fundamentally misaligned with the end-goal of retrieval. Retrieval capabilities emerge only incidentally.
> - **Our Design:** The PSLM objective is a direct response to this. We designed a self-supervised task that is aligned with retrieval by forcing the model to match prefixes (analogs for queries) with their corresponding suffixes (analogs for documents).
> - **The Goal:** This approach aims to move the field from a "BERT" era (requiring heavy supervised fine-tuning) toward a "GPT" era for retrieval, where strong zero-shot capabilities can emerge directly from scalable, self-supervised pretraining.
>
> We believe this philosophy is clearly articulated as the primary motivation for our work.
>
> > 3. Can we merge phase 2 with phase 3 into one phase?
>
> In principle, one could combine the data from both phases. However, we deliberately followed the multi-stage training paradigm common in the retrieval literature.
> The two phases serve distinct purposes:
> - **Phase 2** is a broad, high-volume supervised adaptation step using \~238M "positives-only" pairs.
> - **Phase 3** is a more targeted fine-tuning step using a much smaller, higher-quality dataset (\~1.6M triplets) that explicitly includes mined hard negatives.
>
> As noted in our analysis, the sheer volume of data in Phase 2 (two orders of magnitude more than Phase 3) appears to provide the bulk of the supervised performance gain. Phase 3 provides a smaller, more targeted improvement. While merging them is a valid alternative, our staged approach allows for a clearer analysis of the model's response to different types of supervised data
>
> > 4. How about the performance of the proposed method in the long-context scenarios?
>
> This is an excellent direction for future work. Our current study did not focus on long-context retrieval.
>
> Our Phase 1 pretraining used a max sequence length of 2048 tokens. Adapting the PSLM objective for much longer sequences (e.g., 8k+) is a very interesting research question. One could hypothesize that our prefix-suffix objective is well-suited for this, as the "prefix" would need to find its "suffix" in a much larger context of potential negatives. However, it was beyond the scope of our current study.

---

> > ### Comment · Reviewer_PCeL · 2025-11-22
> >
> > Thanks for your responses. Considering the scalability experiments, I will maintain my scores.

---

### Meta-Review · Area_Chair_iWn9 · 2026-01-11

**Summary:**

This paper proposes Prefix-Suffix Language Modeling (PSLM), a self-supervised pretraining approach for retrieval models that aligns prefix and suffix embeddings via contrastive loss to address the misalignment of traditional MLM/CLM objectives with retrieval tasks. It achieves scalable training on web-scale corpora and demonstrates improved zero-shot retrieval performance.

The reviewers' key concerns lie in: lack of design philosophy explanation, insufficient scalability results across model sizes, insufficient baselines, etc.

**Reviewer Concerns:**

Addressed Concerns
- Lack of design philosophy explanation

- Discrepancies in method figures

- Inadequate related work discussion (partial, new references to be added)

- Incorrect IFEval dataset citation

Outstanding Concerns

- Insufficient scalability results across model sizes

- Unreliable performance scaling with model parameters; unsupported scalability claims

- Limited conceptual novelty vs prior contrastive methods

- Insufficient baselines (RetroMAE, E5, BGE etc.)

**Reviewer Scores:**

- Reviewer PCeL: Maintain the original score (6)
- Reviewer 8aH2: likely to maintain the original score (4) despite author clarifications.
- Reviewer rubc:  likely to maintain the original score (2: reject) as key concerns were not fully resolved.
- Reviewer cSe9: likely to keep the original score (4) since scalability and baseline concerns remain.

---

### Decision · Program_Chairs · 2026-01-26

Reject